# Dynamic modes of Notch transcription hubs conferring memory and stochastic activation revealed by live imaging the co-activator Mastermind

F Javier DeHaro-Arbona[1], Charalambos Roussos[1], Sarah Baloul[1], Jonathan Townson[1], María J Gómez Lamarca[1,2], Sarah Bray[1]*

[1]Department of Physiology Development and Neuroscience, University of Cambridge, Cambridge, United Kingdom; [2]Instituto de Biomedicina de Sevilla (IBiS), Hospital Universitario Virgen del Rocıo/CSIC/Universidad de Sevilla, Departamento de Biologıa Celular, Seville, Spain

*For correspondence:
sjb32@cam.ac.uk

Competing interest: The authors declare that no competing interests exist.

**Abstract** Developmental programming involves the accurate conversion of signalling levels and dynamics to transcriptional outputs. The transcriptional relay in the Notch pathway relies on nuclear complexes containing the co-activator Mastermind (Mam). By tracking these complexes in real time, we reveal that they promote the formation of a dynamic transcription hub in Notch ON nuclei which concentrates key factors including the Mediator CDK module. The composition of the hub is labile and persists after Notch withdrawal conferring a memory that enables rapid reformation. Surprisingly, only a third of Notch ON hubs progress to a state with nascent transcription, which correlates with polymerase II and core Mediator recruitment. This probability is increased by a second signal. The discovery that target-gene transcription is probabilistic has far-reaching implications because it implies that stochastic differences in Notch pathway output can arise downstream of receptor activation.

## eLife assessment

This **fundamental** study advances our understanding of how Notch signalling activates transcription by analysing the dynamics of the Mastermind transcriptional co-activator and its role in the activation complex. The evidence is **compelling** and based on state-of-the-art methods with precise quantitative measurements.

## Introduction

Cells face the challenge of transmitting information accurately, so that cell-surface signals are translated into correct transcriptional responses, and how this is achieved mechanistically remains a major question. Notch is a key signalling pathway that leads to gene activation when ligand and receptor engage upon cell contact (*Bray, 2016*; *Kopan, 2012*; *Borggrefe and Oswald, 2009*). The physical interaction brings about a conformational change that permits proteolytic cleavage and release of Notch intracellular domain (NICD) (*Kovall et al., 2017*; *Kopan and Ilagan, 2009*; *Sprinzak and Blacklow, 2021*). This moiety forms a complex with CSL (CBF-1/RBPJ-κ in mammals, Suppressor of Hairless in *Drosophila* and LAG-1 in *Caenorhabditis elegans*), a transcription factor that binds to specific DNA motifs, and Mastermind (Mam), a coactivator (; *Nam et al., 2006*; *Wilson and Kovall, 2006*). This tripartite activation complex promotes transcription from the target genes where it is

**eLife digest** To correctly give rise to future tissues, cells in an embryo must receive and respond to the right signals, at the right time, in the right way. This involves genes being switched on quickly, with cells often ensuring that a range of molecular actors physically come together at 'transcription hubs' in the nucleus – the compartment that houses genetic information. These hubs are thought to foster a microenvironment that facilitates the assembly of the machinery that will activate and copy the required genes into messenger RNA molecules. The resulting 'mRNAs' act as templates for producing the corresponding proteins, allowing cells to adequately respond to signals.

For example, the activation at the cell surface of a molecule called Notch triggers a series of events that lead to important developmental genes being transcribed within minutes. This process involves a dedicated group of proteins, known as Notch nuclear complexes, quickly getting together in the nucleus and interacting with the transcriptional machinery. How they do this efficiently at the right gene locations is, however, still poorly understood. In particular, it remained unclear whether Notch nuclear complexes participate in the formation of transcription hubs, as well as how these influence mRNA production and the way cells 'remember' having been exposed to Notch activity.

To investigate these questions, DeHaro-Arbona et al. genetically engineered fruit flies so that their Notch nuclear complexes and Notch target genes both carried visible tags that could be tracked in living cells in real time. Microscopy imaging of fly tissues revealed that, due to their characteristics, Notch complexes clustered with the transcription machinery and formed transcription hubs near their target genes.

All cells exposed to Notch exhibited these hubs, but only a third produced the mRNAs associated with Notch target genes; adding a second signal (an insect hormone) significantly increased the proportion. This illustrates how 'chance' and collaboration influence the way the organism responds to Notch signalling. Finally, the experiments revealed that the hubs persisted for at least a day after removing the Notch signal. This 'molecular memory' led to cells responding faster when presented with Notch activity again.

The work by DeHaro-Arbona sheds light on how individual cells respond to Notch signalling, and the factors that influence the activation of its target genes. This knowledge may prove useful when trying to better understand diseases in which this pathway is implicated, such as cancer.

recruited. The sites of recruitment differ according to the cellular context, resulting in different transcriptional outcomes and suggesting that other factors are important in preparing the targets for activation (*Bray and Gomez-Lamarca, 2018*). In addition, release of NICD brings about rapid and robust transcriptional responses within minutes, raising the question how the molecules of cleaved Notch achieve this so efficiently (*Boukhatmi et al., 2020*; *Housden et al., 2013*; *Ilagan et al., 2011*).

Regulation of transcription must be tightly controlled in space, time, and genomic location (*Cramer, 2019*; *Lee and Young, 2013*). Many different studies report that sequence-specific transcription factors, key co-activators, and RNA polymerase II (Pol II) itself undergo dynamic clustering within a nucleus (*Cho et al., 2022*; *Sabari et al., 2018*; *Cho et al., 2018*; *Rippe and Papantonis, 2022*). Clustering appears to be mediated by a combination of specific structure-mediated interactions (e.g. DNA-binding, protein–protein interactions) and multivalent interactions among intrinsically disordered regions (IDRs) present in most transcription factors (*Brodsky et al., 2020*; *Trojanowski et al., 2022*). In this way, transcription is regulated by the formation of functionally specialised local protein microenvironments or transcription 'hubs' associated with target enhancers (*Demmerle et al., 2023*). In some cases, these have the properties of 'condensates' whose formation and dissolution have been explained by the process of phase separation (*Hnisz et al., 2017*). As the resulting assembly is non-stoichiometric, it may enable a small number of transcription factor molecules to drive productive transcription. Such a mechanism could thus explain how NICD, whose nuclear levels are frequently below the level of detection in vivo, can successfully promote robust-target gene transcription (*Trylinski et al., 2017*). Indeed, all members of the Notch activator complex contain unstructured regions that could contribute to the assembly of a hub.

The formation of the tripartite Notch activator complex involves a conserved helix in the N-terminal region of Mam proteins which is responsible for the direct interactions with CSL and NICD

(*Nam et al., 2006*; *Wilson and Kovall, 2006*). The remainder of the large Mam proteins are poorly conserved and appear to be predominantly unstructured albeit to have potential roles in recruiting other cofactors (*Kitagawa, 2016*; *Just Ribeiro and Wallberg, 2009*). For example, there is evidence that human MAML1 interacts with the histone acetyl transferase CBP/p300, which is present at Notch-regulated enhancers in genome-wide studies (*Just Ribeiro and Wallberg, 2009*; *Baghdadi et al., 2018*; *Castel et al., 2013*), and whose recruitment is implicated in activating some targets (*Rogers et al., 2020*; *Fryer et al., 2002*). The C-terminal portion of MAML1 is also suggested to recruit CDK8, the enzymatic core of the Mediator kinase module (*Fryer et al., 2004*; *Janody and Treisman, 2011*; *Wallberg et al., 2002*). What role these factors play in the recruitment, dynamics, and assembly of functional Notch transcription assemblies, and whether these acquire hub-like properties, is unclear.

Live imaging of endogenously tagged proteins offers a non-invasive approach to probe the assembly and composition of transcription hubs in vivo. Using this strategy, we have previously shown that CSL is recruited in a very dynamic manner to a target genomic locus in vivo (*Gomez-Lamarca et al., 2018*). However, as CSL exists in both co-repressor and co-activator complexes (*Franz and Kovall, 2018*), the extent that these dynamics reflect the characteristics of the activation complex remains to be established. Here we incorporated fluorescent tags into the endogenous Mam protein to investigate the behaviours of this Notch nuclear co-activator in vivo, in combination with a method for live imaging of a target genomic locus that responds robustly to Notch activation. The emerging model is that Notch activity leads to the formation of 'transcription hubs' that exist in different states. When Mam is present, key components of the transcriptional machinery become locally concentrated but, surprisingly, in the absence of synergising factors, the conversion to productive transcription only occurs stochastically. In addition, the open chromatin state that is generated decays slowly after Notch withdrawal, providing a memory that enables a more rapid response to a subsequent round of Notch activation.

## Results
### Dynamics of Mastermind recruitment in relation to its partner CSL

The Mam co-activator is an integral part of the Notch transcription complex (*Nam et al., 2006*; *Wilson and Kovall, 2006*; *Fryer et al., 2002*; *Petcherski and Kimble, 2000*). To track Mam dynamic behaviours in vivo, we inserted GFP or Halo into the N-terminus of the endogenous Mam using CRISPR-Cas9 genome editing (*Gomez-Lamarca et al., 2018*). The resulting flies are homozygous viable with no evident phenotypes, indicating that the tagged Mam proteins are fully functional. We first set out to compare the recruitment and dynamics of CSL and Mam at a target locus in Notch OFF and Notch ON conditions, taking advantage of *Drosophila* salivary glands, where the polytene (multiple copy) chromosomes aid detection of chromatin-associated complexes. We used the Int/ParB system, where fluorescently labelled ParB proteins bind to inserted Int sequences, to detect the well-characterised *Enhancer of split complex* [*E(spl)-C*], which contains multiple Notch-regulated genes (*Figure 1—figure supplement 1A*; *Gomez-Lamarca et al., 2018*). Because of the aligned copies of the genome in the polytene chromosomes, the target genomic locus appears as an easily distinguishable fluorescent 'band' in each nucleus during live imaging (*Figure 1A and B*).

The Notch pathway is normally inactive in salivary glands, providing a baseline Notch OFF condition. This was converted to Notch ON by the expression of a constitutively active form of Notch, NΔECD, using the GAL4-UAS system which allows tissue-specific and temporal control. Because it lacks the extracellular domain, NΔECD is constitutively cleaved by gamma-secretase to release NICD, mimicking ligand-induced activation (*Rebay et al., 1993*; *Fortini et al., 1993*; *Struhl and Adachi, 2000*). Comparing the localisation of GFP::Mam in Notch OFF and Notch ON conditions, it was immediately evident that Mam was robustly recruited to *E(spl)-C* in Notch ON conditions in a similar manner to CSL (*Gomez-Lamarca et al., 2018*). In Notch OFF conditions, both proteins were diffuse throughout the nucleus, with a low level of CSL, but not Mam, present at *E(spl)-C*. In Notch ON conditions, strong enrichment of both Mam and CSL was consistently detected around *E(spl)-C*, where the two proteins co-localise in a correlated manner (*Figure 1B*, *Figure 1—figure supplement 1C*).

To compare the dynamics of Mam and CSL at *E(spl)-C* in Notch ON conditions, we performed fluorescence recovery after photobleaching (FRAP) focused on the region defined by the locus-tag. Unlike CSL, which had a rapid recovery ($t^{1/2}$ = 9 s), Mam exhibited slower dynamics ($t^{1/2}$ = 40 s) and failed

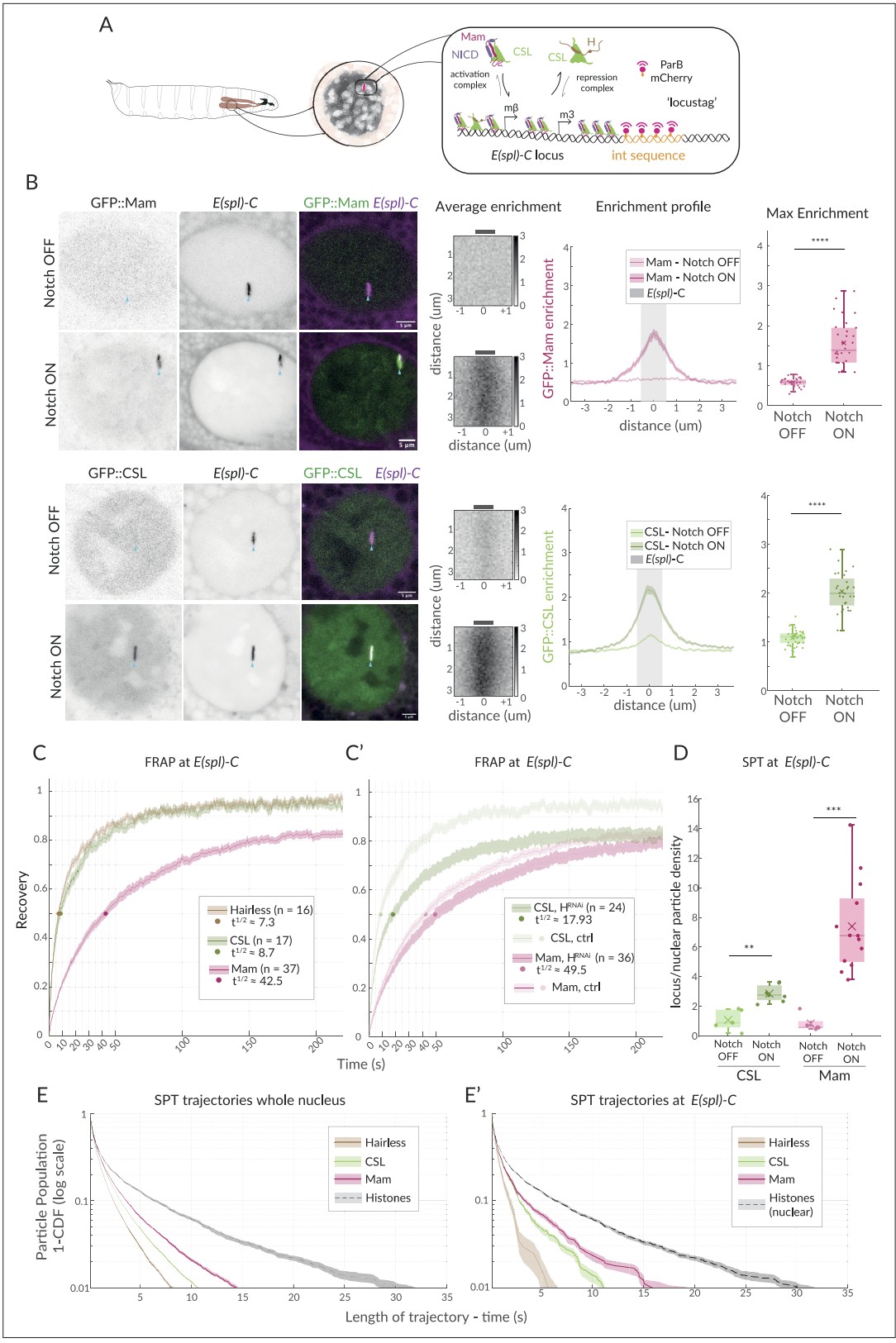

**Figure 1.** Enrichment and dynamics of Mastermind at *E(spl)-C*. (**A**) Schematic overview of live imaging system used. Salivary glands from *Drosophila* larvae (left) have large nuclei with polytene chromosomes (centre, grey shading) in which *E(spl)-C* locus is detected as a band (centre, red) by live imaging when labelled using Int (orange)/ParB (red) system (right and *Figure 1—figure supplement 1A*). Recruitment of activation complexes,

*Figure 1 continued on next page*

*Figure 1 continued*

(CSL [green], NICD [purple], and Mastermind [magenta]) and of co-repression complexes (CSL and Hairless [brown]) is measured by their colocalisation with *E(spl)-C*. (**B**) Live imaging of GFP::Mam and GFP::CSL as indicated in relation to *E(spl)-C* marked by Int-ParB (magenta) in nuclei from Notch OFF (*1151Gal4; UAS-LacZ*) and Notch ON (*1151Gal4; UAS-NΔECD*) salivary glands. CSL and Mam are enriched at *E(spl)-C* in Notch ON but not Notch OFF cells. Average enrichment: each pixel represents average enrichment of all aligned images, centred on *E(spl)-C* locus (0, grey bar). Enrichment profile: mean enrichment, with SEM, plotted on x-axis relative to position, y axis, centred on *E(spl)-C* (0). Grey area indicates region used for max enrichment. Max enrichment: mean of 10 pixels centred on *E(spl)-C* (*Figure 1—figure supplement 1B*) (Mam OFF, n = 32, Mam OFF, n = 30; CSL OFF, n = 45, CSL ON, n = 28; for p-values, see *Supplementary file 1—Table 4*). Box encompasses range between 0.25 and 0.75 quantile, whiskers extend to furthest points not considered outliers, bar marks median, cross marks mean, and each dot is the value for one nucleus. Scale bars represent 5 μm. Genetic combinations for all figures are provided in *Supplementary file 1—Table 3*. (**C, C'**) Dynamics of CSL::GFP (green), GFP::Mam (magenta), and Hairless::GFP (brown) at *E(spl)-C* in Notch ON cells measured by fluorescence recovery after photobleaching (FRAP). Recovery of the indicated proteins was measured in a point bleached region-of-interest centred on *E(spl)*-C and normalised by using another region and efficiency of bleaching. Dots indicate 50% recovery. Legend summarises numbers of nuclei (n) and time to 50% recovery (t1/2). Error represents the standard error of the mean (SEM). (**C'**) FRAP analysis of CSL and Mam in cells depleted for Hairless (*1151Gal4; UAS-Hairless RNAi*) controls from (**C**) are included for comparison. (**D**) Trajectory density at *E(spl)-C* relative to whole nucleus from SPT of CSL::Halo (green) and Halo::Mam (pink) in Notch OFF and Notch ON cells. Both CSL and Mam are significantly enriched in Notch ON. Box plots as in (**B**). CSL-OFF, n = 5, CSL-OFF, n = 7; Mam-OFF, n = 5, Mam-ON, n = 13; for number of trajectories, see *Figure 1—figure supplement 1F* and for p-values see *Supplementary file 1—Table 4*. (**E, E'**) Survival curves depicting duration of trajectories in SPT of the indicated Halo fusion proteins in whole nuclei (**E**) and at *E(spl)-C* (**E'**), in Notch ON cells. Whole nuclei H2B::Halo trajectories are included for comparison in both graphs. For number of trajectories, see *Figure 1—figure supplement 1F*. Error bars represent 95% confidence intervals, obtained from bootstrapping with 100 resampled datasets (*Efron and Tibshirani, 1994*).

The online version of this article includes the following figure supplement(s) for figure 1:

**Figure supplement 1.** Image analysis and effect of Hairless depletion on CSL and Mam.

---

to fully recover over the time course of the experiment (*Figure 1C*). Slower recovery can arise from higher proportion of bound molecules, longer residence times, and/or slower diffusion coefficients (*Wachsmuth, 2014*). The results therefore suggest that the Mam-containing activation complexes have different properties from the majority of CSL complexes.

The other main partner for CSL in *Drosophila* is the co-repressor Hairless. In FRAP experiments, Hairless has a fast recovery, with a profile close to that of CSL (*Figure 1C*), consistent with a significant fraction of CSL being complexed with Hairless even in Notch ON conditions (*Gomez-Lamarca et al., 2018*). The difference in dynamics between CSL and Mam could therefore be explained by the former being involved in two different complexes with different dynamics. To test this, we depleted Hairless using RNAi-mediated knock-down, which we validated by RT-qPCR, and measured the effects on enrichment and dynamics of CSL and Mam (*Figure 1—figure supplement 1D*). As expected, Mam recruitment levels and the dynamics measured by FRAP were unchanged (*Figure 1C'*, *Figure 1—figure supplement 1E*). In contrast, CSL levels and recruitment were reduced and its FRAP recovery was slowed, albeit not to the extent that it recapitulated the Mam profile (*Figure 1C'*, *Figure 1—figure supplement 1E*). Together, the results indicate that the activation complexes, containing CSL and Mam, have slower dynamics than the repressor complexes, containing CSL and Hairless, and that the recovery of CSL reflects its participation in the two types of complexes.

To further investigate the dynamics of Mam and CSL complexes, we performed single-particle tracking (SPT) by sparse labelling of endogenous Halo::Mam and Halo::CSL in live tissue (*Liu et al., 2018*; *Baloul et al., 2024*). Gaussian fitting-based localisation and multiple hypothesis tracking were used for detection and tracking of single particles within the nucleus with an ~20 nm precision (*Gomez-Lamarca et al., 2018*). Using a Bayesian treatment of Hidden Markov Models, vbSPT (*Persson et al., 2013*), trajectories were assigned into two states, defined by a Brownian motion diffusion coefficient, that correspond to 'bound' chromatin-associated molecules (diffusion coefficient 0.01 μm²/s) versus more freely diffusing complexes (diffusion coefficient > 0.25 μm²/s). A greater proportion (55%) of Mam complexes were in the bound state than CSL complexes (39%), consistent with the differences between their FRAP curves (*Figure 1—figure supplement 1F*). We also analysed

the density of particle trajectories in relation to the *E(spl)-C* locus in Notch OFF and Notch ON conditions. In comparison to their average distribution across the nucleus, both CSL and Mam trajectories were significantly enriched in a region of approximately 0.5 µm around the target locus in Notch ON conditions, reflecting robust Notch-dependent recruitment to this gene complex (*Figure 1D*).

To assess whether Mam complexes have longer residence times once recruited to the chromatin, we analysed the duration of trajectories for Mam, CSL, and Hairless. Long trajectories correlate to bound complexes because faster moving particles are rapidly lost from the field of view, and the length of time they are detectable is an indication of relative residence times. There were clear differences between the trajectory durations for Mam, CSL, and Hairless. Mam trajectories had the longest durations (up to 15 s), Hairless trajectories were the shortest (up to 5–7 s), and CSL trajectories were intermediate (up to 10 s) (*Figure 1E*). The differences were recapitulated when only the trajectories in the region around *E(spl)-C* were analysed (*Figure 1E'*). The residences are likely an underestimation because bleaching and other technical limitations also affect the track durations (*Mazza et al., 2012*). Nevertheless, these data confirm that Mam-containing complexes have on average longer residence times than other CSL complexes which, together with the higher proportion of bound molecules overall, explains the slower recovery dynamics measured by FRAP.

The fact that CSL dynamics, measured by FRAP and SPT, are intermediate between Hairless and Mam fit well with it being present in two types of complexes (co-activator [Mam] and co-repressor [Hairless]). However, whether the contribution from CSL co-repressor complexes can fully explain all the observed differences between CSL and Mam is not fully clear. First, when we depleted Hairless, so that the majority of CSL present would be in co-activator complexes, CSL FRAP recovery curves were still substantially different from those of Mam. Second, none of the CSL trajectories had a duration approximating those of the longest-lasting Mam trajectories, despite that over 50,000 CSL trajectories were tracked (compared to 14,000 Mam, *Figure 1—figure supplement 1F*). It is, therefore, possible that, once recruited, Mam can be retained at target loci independently of CSL by interactions with other factors so that it resides for longer.

## Hub-like properties of CSL-Mastermind complexes in Notch active cells

The enrichment of CSL and Mam around *E(spl)-C* that we detect by live imaging is not unexpected because this locus has multiple genes containing CSL binding-motifs (*Gomez-Lamarca et al., 2018*; *Krejcí and Bray, 2007*). However, the diffuse enrichment was not maintained when the tissues were fixed, a characteristic reported for proteins present in condensate-like hubs (*Irgen-Gioro et al., 2022*; *Figure 2—figure supplement 1A*). The localised concentration of exchanging CSL and Mam complexes around *E(spl)-C* in Notch ON nuclei may therefore have properties of a transcription hub, with some of the recruitment being reliant on weak interactions mediated by low-complexity regions (*Boija et al., 2018*; *Sönmezer et al., 2021*; *Tsai et al., 2019*).

First, we asked to what extent CSL recruitment was correlated with the number of CSL binding motifs. To do so, we took advantage of fly strains containing multiple copies of CSL motifs inserted at an ectopic position in the genome (*Kuang et al., 2021*) and compared the recruitment with insertions containing 12 or 48 CSL motifs. In Notch ON conditions, these insertions were sufficient to generate an ectopic band of CSL recruitment similar to the native *E(spl)-C* (*Figure 2A*). Remarkably the amounts of CSL recruited to loci with 12 and 48 ectopic sites were almost identical. Thus, there is not a direct correlation between the number of motifs and the amount of CSL recruited under these conditions, although we note that DNA binding is a prerequisite as no recruitment occurs with CSL mutant that lacks DNA binding (*Gomez-Lamarca et al., 2018*). Nor does it appear that the amounts of CSL complexes are a limiting factor as there was no decrease in recruitment at the endogenous *E(spl)-C* locus, even in nuclei with a 48 CSL-motif insertion.

Second, we questioned whether the non-stochiometric recruitment of activator complexes to loci with CSL motifs might involve additional weak protein–protein interactions between some components, as observed in several transcription hubs where intrinsically disordered protein domains play a part (*Brodsky et al., 2020*; *Kawasaki and Fukaya, 2023*). We used IDR prediction algorithms (*Emenecker et al., 2021*) to identify IDRs within NICD, CSL, Mam, and Mediator1 (Med1, part of the mediator complex) (*Figure 2B*) and generated transgenic flies where each IDR was expressed as a fusion with GFP. The recruitment of the IDR::GFP fusions to *E(spl)-C* was then measured in Notch ON and OFF conditions (*Figure 2C*). Of the IDRs tested, the C-terminal region from NICD was the most

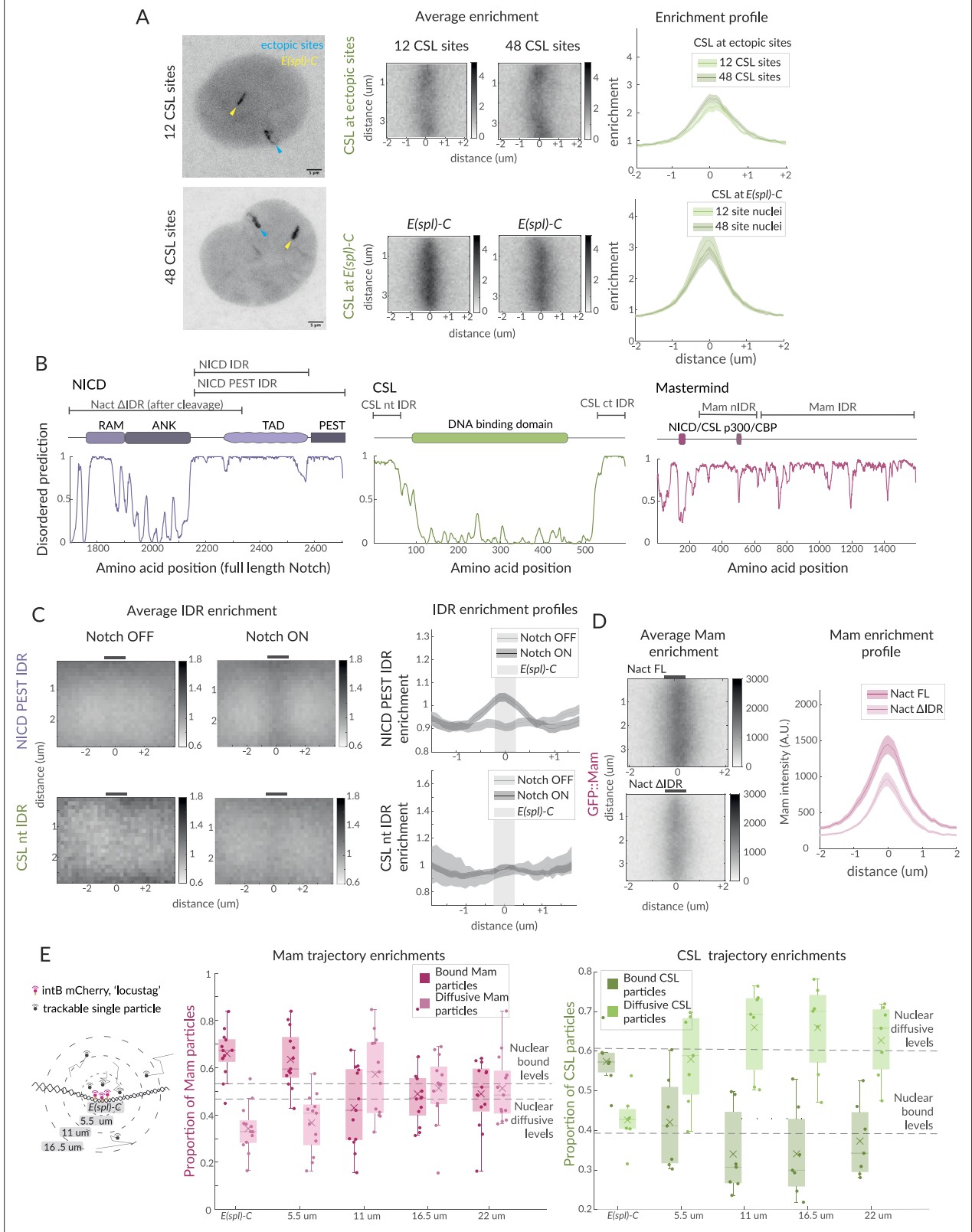

**Figure 2.** Hub-like properties of recruited complexes in Notch ON nuclei. (**A**) Representative images of GFP::CSL in Notch ON nuclei containing an ectopic array of 12 or 48 CSL binding sites. GFP::CSL is recruited to *E(spl)-C* (yellow) and the ectopic array (blue). Average enrichment, enrichment profile as in *Figure 1B* (ectopic array 12, n = 45, ectopic array 48, n = 40; *E(spl)-C* 12, n = 45, *E(spl)-C* 48, n = 40). (**B**) Domain organisation of the indicated proteins is diagrammed above the prediction scores (Metapredict V2) of protein disorder for each. Regions tested as intrinsically disordered

*Figure 2 continued on next page*

*Figure 2 continued*

regions (IDRs) are indicated. (**C**) Average enrichments and enrichment profiles of NICD-PEST IDR (Notch OFF, n = 62, Notch ON, n = 67) and CSL-nt IDR (Wilcoxon rank sum test: Notch OFF, n = 8, Notch ON, n = 40) at *E(spl)-C* in Notch OFF and Notch ON cells. (**D**) Average enrichments and enrichment profiles (as in *Figure 1B*) of GFP::Mam at *E(spl)-C* in nuclei expressing intact (Nact) and IDR deleted (Nact ΔIDR) Notch constructs (Nact, n = 58, Nact ΔIDR, n = 55). p-Values in *Supplementary file 1—Table 4*. (**E**) Diagram illustrates concentric ring analysis of single-particle trajectories at increasing distances from *E(spl)-C*. Graphs plot proportions of bound (dark shading) and diffusive (light shading) particles of Halo::Mam or Halo::CSL within each ring. Dashed lines indicate nuclear proportions of each population as indicated. For the number of trajectories, see *Figure 1—figure supplement 1F*.

The online version of this article includes the following figure supplement(s) for figure 2:

**Figure supplement 1.** Notch activation complexes exhibit hub-like properties.

strongly enriched at *E(spl)-C* in Notch ON conditions. A low level of enrichment was also evident for a shorter NICD fragment, which lacked the C-terminal PEST sequences and for Med 1 IDR but not for the terminal regions of CSL. Surprisingly, little or no enrichment occurred with Mam-IDRs, despite this large protein being reported to interact with p300 and other factors (*Figure 2—figure supplement 1C*) and the mammalian MamL1 forming puncta when overexpressed (*Wu et al., 2000*). Notably, however, Mam-nIDR::GFP fusion was present in droplets, suggesting that it can self-associate when present in a high local concentration (*Figure 2—figure supplement 1B*; *Banani et al., 2017*). Our results, therefore, suggest that the IDR in NICD may contribute to the localised enrichments at target loci in Notch ON cells, potentially in combination with IDRs present in other recruited factors. In support of this hypothesis, deletion of the IDR from NICD led to a reduction in the levels and stability of Mam recruitment to *E(spl)-C* (*Figure 2D*).

Third, we reasoned that the presence of a transcription hub, where complexes are retained in the vicinity via protein interactions, should result in local changes in the behaviours of CSL and Mam. After segregating the SPT trajectories according to their diffusion properties as described above (*Figure 1—figure supplement 1F*), we analysed the spatial distribution of the slow and fast populations in relation to the *E(spl)-C,* defined as the area within 550 nm of the locus-tag. Based on the shape and centre of this region, concentric zones were defined at 550 nm distances and the proportions of slow- and fast-moving particles in each zone were calculated. The results show that there is an enrichment for slow-diffusing and a depletion of fast-moving particles close to the locus, with these altered properties extending to a region of up to 1 μm away (*Figure 2E*).

Together, our data support the model that CSL-Mam complexes are recruited and form a hub of high protein concentrations around the target locus in Notch ON conditions and suggest that IDR interactions, as well as DNA binding, contribute to their recruitment and retention in a region surrounding the active enhancers.

## Mediator CDK module is required for stable Mastermind recruitment

The hub-like properties and slower turnover of Mam complexes compared to CSL suggest that other factors will be involved in their stabilisation. We, therefore, tested the consequences of inhibiting or depleting different factors to distinguish those required for Mam enrichment at *E(spl)-C* in Notch ON nuclei. We first asked whether active transcription was required for Mam recruitment by exposing the tissue to triptolide, a fast and specific inhibitor of transcription initiation which effectively inhibited transcription in Notch ON cells (*Figure 3—figure supplement 1A*; *Titov et al., 2011*). No change in Mam enrichment or recovery was detected, arguing that Mam recruitment is not dependent on initiation or RNA production (*Figure 3A and B*). As previous studies have reported an interaction between Mam and the histone acetylase CBP/p300 (*Fryer et al., 2002*; *Wallberg et al., 2002*; *Clark et al., 2015*; *Oswald et al., 2001*), we next inhibited CBP activity using a potent and selective inhibitor A485 (*Lasko et al., 2017*). Tissues were exposed to A485 for 1 hr which was sufficient to severely reduce the levels of H3K27Ac and *E(spl)*m3 transcription, indicating that the treatment was effective (*Figure 3—figure supplement 1B and C*). Surprisingly, however, there was no change in the recruitment of Mam in these conditions (*Figure 3C*). We also depleted CBP by RNAi and, as with the drug treatment, Mam recruitment was unchanged (*Figure 3D*, *Figure 3—figure supplement 1J*). Based on these results, we conclude that CBP is not essential for the recruitment of Mam complexes to the hub formed at the *E(spl)-C* locus.

As several studies have suggested that the Mediator CDK module, containing Med12, Med13, CDK8, and CycC, influences Notch-dependent transcription, we next focused on the role of this

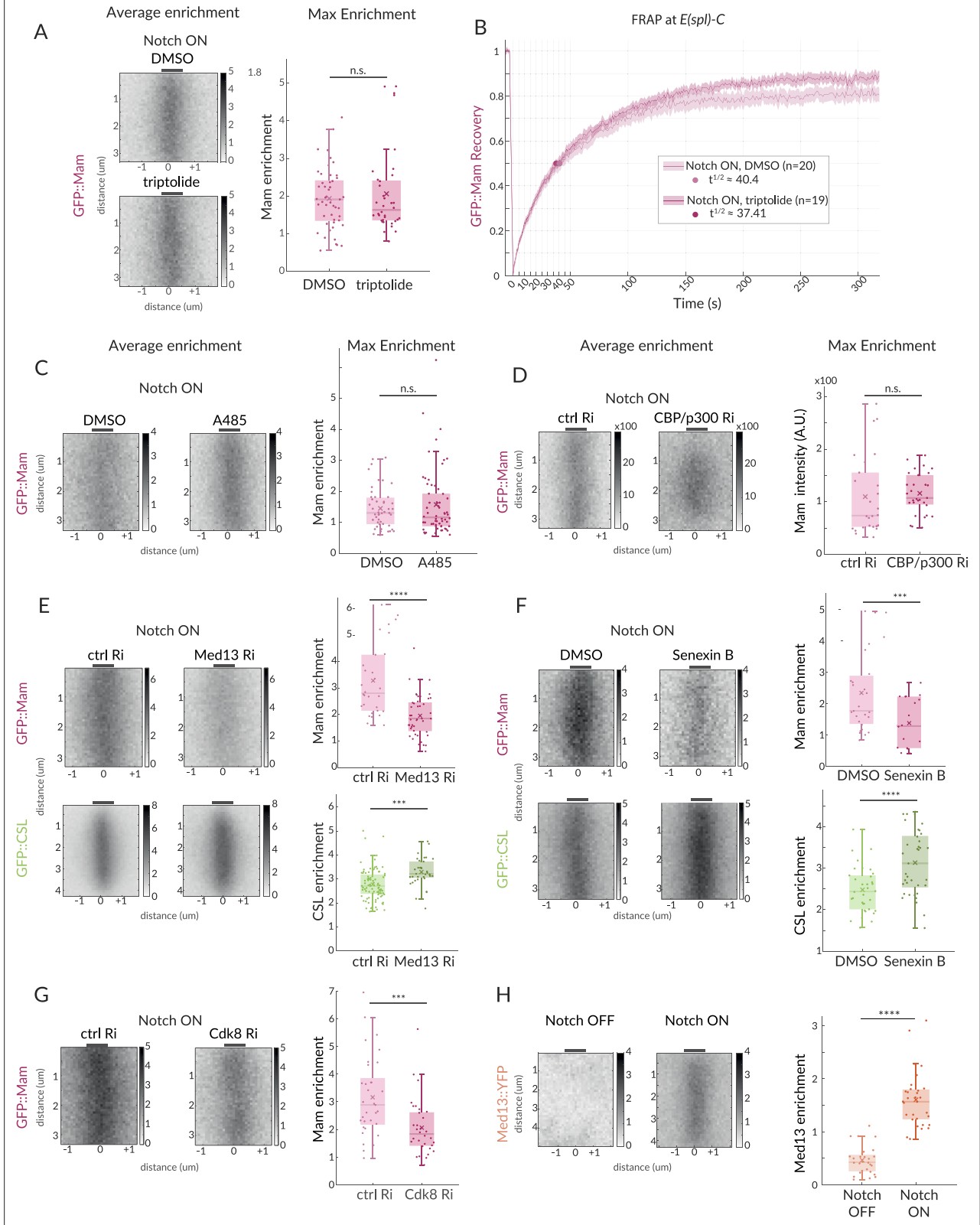

**Figure 3.** Mam enrichment requires Mediator CDK module but is independent of transcription and CBP/p300. (**A, B**) GFP::Mam recruitment levels and dynamics at *E(spl)-C* in Notch ON tissues treated with triptolide or DMSO as a control. (**A**) Average and Max enrichment as in *Figure 1B* (DMSO n = 49, triptolide, n = 36). (**B**) FRAP recovery curves, plotted as in *Figure 1C*. (**C–F**) Average enrichment and Max enrichment of GFP::Mam and GFP::CSL, as indicated, at *E(spl)-C* in Notch ON control and treated tissues. (**C, D**) No change in recruitment following inhibition (**C**, A485) or genetic knockdown (**D**,

*Figure 3 continued*

RNAi) of CBP/p300 compared to control, DMSO and *yellow* RNAi (yRi), respectively (DMSO, n = 47, A485, n = 62; *y* RNAi, n = 26, CBP/p300 RNAi, n = 31). (**E, F**) Reduced recruitment of GFP::Mam (magenta) but not CSL::GFP (green) following (**E**) genetic knockdown of Med13 and (**F**) inhibition of CDK8 with Senexin B (E: Mam control, n = 30, Mam Med13Ri, n = 45; CSL control, n = 92, CSL Med13Ri, n = 36); (**F**: Mam ctrl n = 27, Mam Senexin, n = 43; CSL ctrl, n = 34, CSL Senexin, n = 37). (**G**) Genetic knockdown of Cdk8 results in decreased Mam recruitment compared to controls (ctrl, n = 33, Cdk8 Ri, n = 32). Box plots in (**A–F**) as in *Figure 1B*. For p-values, see *Supplementary file 1—Table 4*. (**H**) Average enrichment and max enrichment of Med13::YFP at *E(spl)-C* in Notch OFF, n = 28 and Notch ON, n = 31; for p-values, see *Supplementary file 1—Table 4*.

The online version of this article includes the following figure supplement(s) for figure 3:

**Figure supplement 1.** Mediator complex, but not CBP/p300, is involved in Mam enrichment.

complex (*Fryer et al., 2004*; *Janody and Treisman, 2011*; *Kuang et al., 2020*). First, we depleted Med13 and CDK8 using well-validated RNAi lines (*Dobi et al., 2014*; *Li et al., 2020*, *Figure 3— figure supplement 1J*). Med13 depletion led to a loss of *E(spl)*m3 transcription, confirming its role (*Figure 3—figure supplement 1E*). In these conditions, we detected a substantial reduction in Mam recruitment to *E(spl)-C* in Notch ON cells, suggesting that the CDK module plays a role in retaining Mam at target sites (*Figure 3E and G*). Second, we incubated the tissues in Senexin B or Senexin A, two drugs that target CDK8 activity (*Roninson et al., 2016*; *McDermott et al., 2017*) for 1 hr prior to imaging. In both cases, the treatment was sufficient to significantly impair Mam levels at *E(spl)*-C (*Figure 3F*, *Figure 3—figure supplement 1D*) while a CDK9 inhibitor, NVP2, had no effect on the same assays despite a loss of *E(spl)*m3 transcription (*Figure 3—figure supplement 1H and I*). We note that human MamL1 is a high-confidence CDK8 target (*Poss et al., 2016*), but as the phospho-sites are not conserved it is unclear whether Mam would also be a direct target of the kinase. Third, we investigated whether CDK module was recruited to *E(spl)-C*, in Notch ON nuclei using an existing line in which YFP is fused to Med13 (*Lye et al., 2014*; *Lowe et al., 2014*). Significant enrichment of Med13::YFP was detected at *E(spl)-C* in Notch ON nuclei (*Figure 3H*), demonstrating that it is present in the hub with Mam.

Despite their effects on Mam recruitment, neither depletion of Med13 nor Senexin treatments caused any significant reduction in the levels of CSL recruited. On the contrary, a small increase occurred (*Figure 3E and F*). One explanation for the differences in the effects on CSL and Mam could be that there is increased recruitment of CSL complexes containing the co-repressor, Hairless, following Senexin treatment. We, therefore, monitored Hairless recruitment under these conditions but detected no increase in Senexin-treated nuclei (*Figure 3—figure supplement 1F*). However, the overall nuclear levels of Hairless are quite high, making it difficult to detect and quantify any changes at *E(spl)-C*. We, therefore, investigated the impact of Senexin on recruitment of a mutant CSL (CSL-H[mut]) with compromised Hairless binding. This retains normal recruitment in control Notch ON conditions (*Figure 3—figure supplement 1G*), but, strikingly, it is not enriched at *E(spl)-C* in Senexin-treated tissues. This result suggests that CSL requires Hairless interaction for its recruitment to *E(spl)-C* upon CDK8 inhibition. The differences in the effects on Mam and CSL imply that the CDK module is specifically involved in retaining Mam in the hub, and that in its absence other CSL complexes containing Hairless 'win-out' either because the altered conditions favour them and/or because they are the more abundant.

## Mastermind is not essential for chromatin accessibility

The observed differences between CSL and Mam in their dynamics and dependency on the CDK module led us to speculate that they make different functional contributions to the transcription hub. To probe the role of Mam, we examined the consequences when its recruitment to the complex was prevented. The N-terminal peptide from Mam functions as a dominant negative (MamDN) by occupying the groove formed by CSL-NICD and, when overexpressed, prevents transcription of target genes (*Nam et al., 2006*; *Wilson and Kovall, 2006*; *Helms et al., 1999*.) As predicted, MamDN expression in Notch ON conditions prevented recruitment of full-length Mam to *E(spl)-C* (*Figure 4A*). To verify that MamDN also inhibited transcription under these conditions, we used single-molecule fluorescent in situ hybridisation (smFISH) with probes targeting *E(spl)-C* transcripts that are robustly upregulated in this tissue (*Gomez-Lamarca et al., 2018*). Nascent transcripts of *E(spl)*m3 were detected at *E(spl)-C* in Notch ON nuclei and were at very reduced levels in nuclei co-expressing MamDN (*Figure 4D and D'*), confirming that target-gene transcription is blocked.

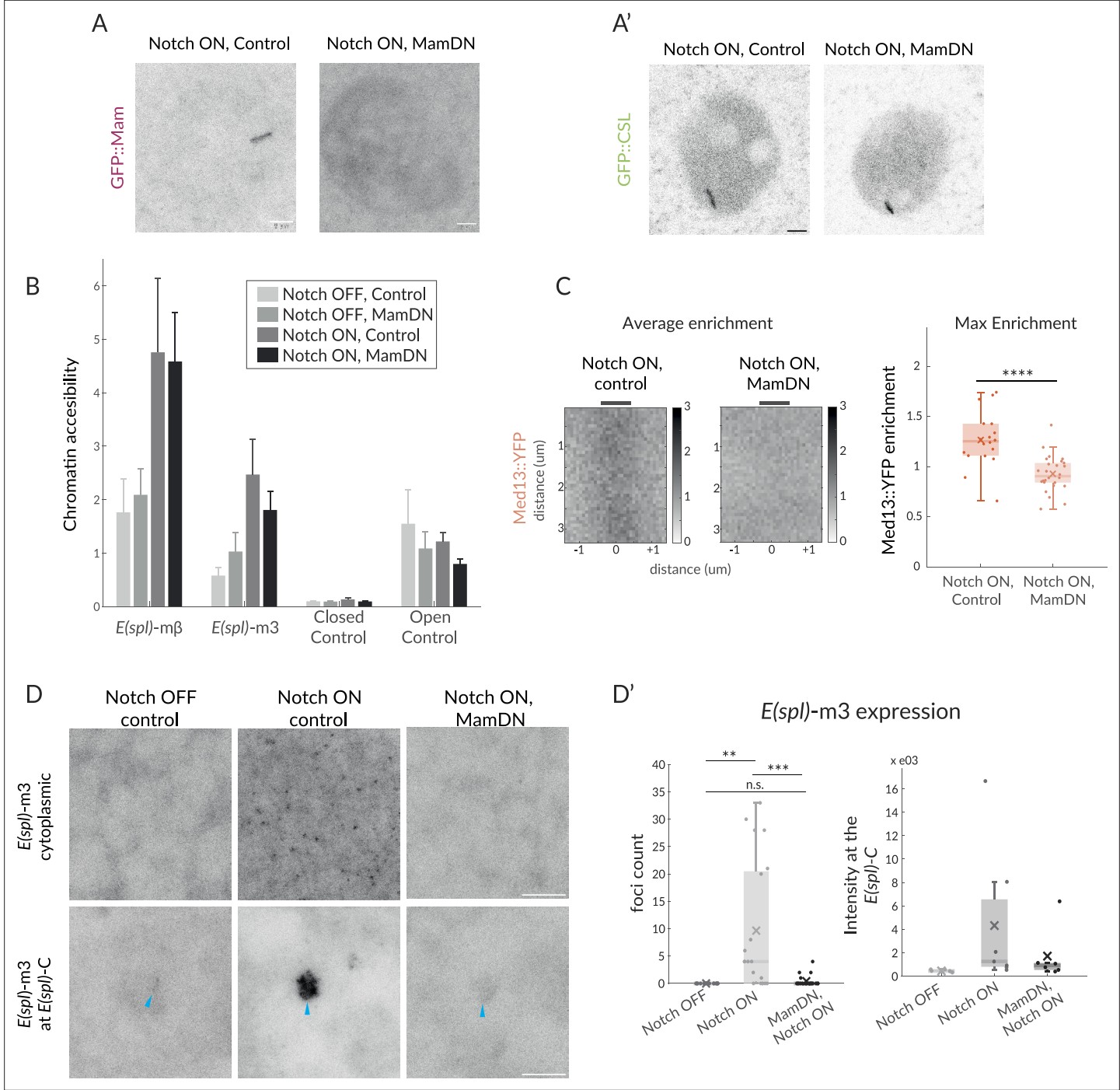

**Figure 4.** Mam is not necessary for CSL enrichment and chromatin opening but necessary for transcription and Mediator 13 enrichment. (**A–A'**) Recruitment of GFP::Mam in Notch ON nuclei is perturbed by MamDN (*1151-Gal4 UAS-MamDN*) (**A**) while recruitment of GFP::CSL is unaffected (**A'**). Control UAS was included in Notch ON combinations, details of genotypes are provided in ***Supplementary file 1—Table 3***. Scale bars represent 5 µm. (**B–B'**) Accessibility of Notch-regulated enhancers adjacent to *E(spl)* -mβ and *E(spl)* -m3 under the conditions indicated was probed using ATAC-qPCR. Values were normalised to Rab11, error bars represent SEM. Closed control is a noncoding genomic region in chromosome 3 and open control' Eip78, an ecdysone responsive region active in L3 larval stage. (**C**) Average enrichment and max enrichment of Med13::YFP at *E(spl)-C* in Notch ON conditions only, and in combination with MamDN. Quantifications as in ***Figure 1B***, OFF, n = 28, ON, n = 31. For p-values, see ***Supplementary file 1—Table 4***. (**D–D'**) Expression of *E(spl)*-m3, detected by smFISH in the conditions indicated; representative images from cytoplasm (upper; 1.8 mm$^3$) and nucleus (lower; centred on *E(spl)-C*, blue arrows). Scale bars represent 5 µm. Graphs (**D'**), number of RNA puncta (cytoplasmic, n = 18, 20, and 22) and locus intensity (nucleus, n = 6, 7, and 6) in Notch OFF, Notch ON ctrl, and Notch ON MamDN as indicated; boxplots as described in ***Figure 1B***.

The online version of this article includes the following figure supplement(s) for figure 4:

*Figure 4 continued on next page*

*Figure 4 continued*

**Figure supplement 1.** Mam perturbations do not affect CSL or IDR enrichment.

By contrast, recruitment of CSL was unaffected by MamDN expression. It was robustly recruited to *E(spl)*-C at a similar level to untreated Notch ON nuclei despite the absence of full-length Mam (*Figure 4A'*; *Gomez-Lamarca et al., 2018*). Identical results were obtained when Mam was depleted by RNAi (*Figure 4—figure supplement 1A*; *Gomez-Lamarca et al., 2018*). The fact that CSL was still strongly enriched under these conditions argues that some of the Notch-induced changes at *E(spl)-C* occur independently of the co-activator, as proposed previously (*Gomez-Lamarca et al., 2018*). In contrast to CSL, the enrichment of Med13 was lost in the presence of MamDN, revealing that Med13 recruitment requires Mam, as well as the converse (*Figure 4C*).

Since MamDN does not prevent CSL recruitment, although it blocks recruitment of Med13 and transcription, we hypothesised that some Notch-induced effects at target enhancers may not require Mam. One consequence from Notch activation is an increase in chromatin accessibility at sites where CSL is recruited (*Gomez-Lamarca et al., 2018*; *Giaimo et al., 2017*). We, therefore, asked whether changes in chromatin accessibility still occur in the presence of MamDN by performing ATAC and analysing by qPCR the accessibility of the regulatory regions abutting *E(spl)*mβ and *E(spl)*m3 (*Gomez-Lamarca et al., 2018*). Both regions significantly increased in accessibility in Notch ON nuclei which was maintained in the presence of MamDN (*Figure 4B*). In addition, we found that the levels of GFP::NICD-IDR recruited to *E(spl)*-C were similar in the presence and absence of MamDN (*Figure 4—figure supplement 1B*), arguing that a modified hub is still formed.

Altogether these observations support the model that the increased chromatin accessibility elicited by Notch activation can occur independently of Mam and must rely on other functions conferred by CSL and NICD (*Gomez-Lamarca et al., 2018*), while Mam is essential to recruit the Mediator CDK module and enable transcription.

## CSL recruitment and chromatin accessibility persist after Notch inactivation, conferring memory

Our results suggest that there are two or more separable steps involved in forming an active transcription hub in Notch ON cells: a Mam-independent change in chromatin accessibility, a Mam-dependent recruitment of Mediator, and initiation of transcription. If these are discrete steps, we reasoned that they might decay with different kinetics when Notch activity is removed. We took advantage of the thermosensitive Gal4/Gal80$^{ts}$ system to switch off Notch activity and assessed the consequences on CSL and Mam recruitment at two different time points: 4 hr and 8 hr after the switch-off. Imaging Mam::GFP and CSL::mCherry simultaneously, it was evident that Mam recruitment levels decreased more rapidly. CSL remained relatively constant through both time points while, in contrast, the levels of Mam at *E(spl)*-C decreased sharply after 4 hr (*Figure 5A*). Based on these results, we propose that, after Notch activity decays, the locus remains accessible because when Mam-containing complexes are lost they are replaced by other CSL complexes (e.g. co-repressor complexes).

As Notch removal leads to a loss of Mam, but not CSL, from the hub, we hypothesised that it would recapitulate the effects of MamDN on chromatin accessibility and transcription of targets. We, therefore, measured the accessibility of target enhancers at *E(spl)*mβ and *E(spl)*m3 at the 8 hr time point by ATAC. Neither enhancer exhibited any reduction in accessibility at this time point, consistent with the continued recruitment of CSL and the results obtained with MamDN (*Figure 5B*). In contrast, the gradual loss of Mam complexes was accompanied by reduced transcription, as detected by smFISH. Expression of *E(spl)*m3 was already significantly reduced at 4 hr when levels of nascent transcripts at *E(spl)*-C and of cytoplasmic transcripts had both decreased (*Figure 5—figure supplement 1A*). By 8 hr both nascent and cytoplasmic *E(spl)*m3 transcripts were almost undetectable (*Figure 5—figure supplement 1A*). Thus, the changes in transcription correlate well with the reduction in Mam recruitment, whereas the chromatin accessibility persists in the absence of Mam. We next investigated whether CSL remained enriched at the locus for an extended period following Notch inactivation. Indeed, we were still able to detect a strong CSL enrichment 24 hr after transfer to the non-permissive temperature at time when, consistent with shorter OFF periods, there was no detectable transcription of *E(spl)*m3 (*Figure 5C*).

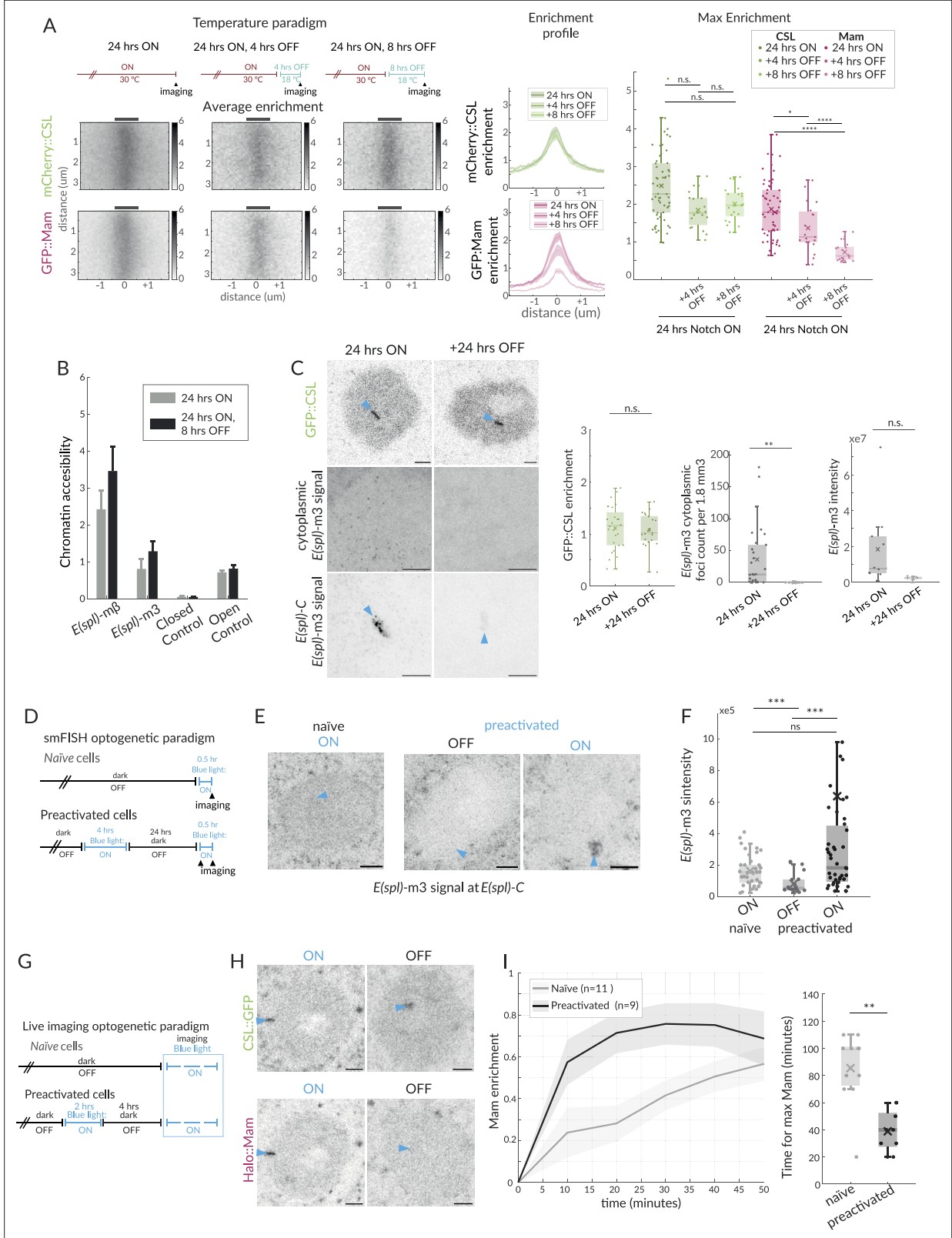

**Figure 5.** Effects of Notch withdrawal on hub composition imply memory state. (**A**) Temperature paradigm: withdrawal of Notch activity using the thermosensitive Gal4/Gal80[ts] system; after a 24 hr ON period, larvae were transferred to 18C (Gal4 inhibited) for the indicated periods (top) and enrichment of mCherry::CSL and GFP::Mam at *E(spl)-C* imaged. Average enrichment, enrichment profiles, and max enrichment quantified as in *Figure 1B*. 24 hr ON, n = 28, 24 hr ON, 4 hr OFF, n = 21, 24 hr ON, 8 hr OFF, n = 19. p-Values are summarised in *Supplementary file 1—Table 4*. (**B**)

*Figure 5 continued on next page*

*Figure 5 continued*

Accessibility of enhancers at *E(spl)* -mβ and *E(spl)* -m3, in Notch ON and after Notch withdrawal (8 hr) was probed using ATAC-qPCR, as in **Figure 4B**. (**C**) Recruitment of CSL::GFP and expression of *E(spl)*-m3, detected by single molecule fluorescent in situ hybridisation (smFISH) in the conditions indicated. Robust recruitment of CSL::GFP at E(spl)-C persists after switching to none-permissive temperature (Notch OFF) for 24 hr, representative image and graph (green) with intensity quantifications for multiple nuclei (24 hr ON, n = 28, +24 hr OFF, n = 21). For smFISH, representative images from cytoplasm (upper; 1.8 mm³) and nucleus (lower; centred on *E(spl)-C*, blue arrow) are shown; scale bars represent 5 μm. RNA puncta are absent after 24 hr Notch OFF. Graphs show the number of cytoplasmic RNA puncta (left) from n = 30, 16 regions and RNA fluorescence intensity at *E(spl)*-C in the conditions indicated from n = 10, 5 nuclei, with boxplots as described before (**Figure 1B**). p-Values in **Supplementary file 1—Table 4**. (**D**) Optogenetic paradigm used for smFISH: Conditions used to switch ON and OFF Notch activity using OptiC Notch (**Townson et al., 2023**). (**E, F**) Expression of *E(spl)*-m3, detected by smFISH in the conditions indicated. (**E**) Representative images from the nucleus (centred on *E(spl)-C*, blue arrows). Scale bars represent 5 μm. (**F**) Graphs for fluorescence intensity at *E(spl)*-C in the conditions indicated from n = 38, 21, and 43 nuclei, with boxplots as described before (**Figure 1B**). p-Values in **Supplementary file 1—Table 4**. (**G**) Optogenetic paradigm used for live imaging: conditions used to switch ON and OFF Notch activity using OptiC Notch (**Townson et al., 2023**). (**H**) Recruitment of CSL::GFP and Mam::Halo at *E(spl)-C*, measured following blue light activation in OptiC Notch expressing tissues for 2 hr (ON) and after 4 hr in dark (OFF). GFP::CSL remains enriched, Halo::Mam is depleted. (**I**) Recruitment of Halo::Mam in preactivated nuclei compared to naïve nuclei; temporal profile of Mam::Halo enrichment (right) and time taken to max enrichment (left). Error bars in the profile represents the SEM, values were normalised from 0 to 1 (similar to FRAP recovery, see **Figure 1** and 'Materials and methods'), n indicates the number of nuclei from >3 salivary glands. Boxplot parameters as in **Figure 1B**; naïve, n = 11, preactivated n = 9.

The online version of this article includes the following figure supplement(s) for figure 5:

**Figure supplement 1.** Notch deactivation leads to loss of *E(spl)*-m3 transcription.

One possible consequence of the prolonged CSL enrichment in Notch OFF conditions is that target loci will retain 'memory' of previous Notch activation that would make them more receptive to a subsequent exposure to Notch activity. To investigate, we took advantage of the temporal control provided by optogenetic release of NICD using OptIC-Notch{ω} (**Townson et al., 2023**). We compared the response to Notch activation in cells that had been 'preactivated' with blue light for 4 hr and subsequently kept in the dark for 24 hr, and 'naïve' cells, which had been kept solely in the dark and hence had no prior Notch activity (**Figure 5D–F**). We confirmed that cells had no residual transcription after being kept in dark, Notch inactive conditions for 24 hr ('OFF'; **Figure 5E and F**). Strikingly, after 30 min in blue light to activate Notch, cells that had been preactivated showed higher levels of *E(spl)*m3 transcription compared to naïve cells, indicating that the previous Notch exposure renders them more sensitive (**Figure 5E and F**). To investigate whether a previous activation also influences Mam recruitment, we tracked its recovery in a live imaging experiment. As a starting point we deployed a 4 hr OFF period, after which Mam was fully depleted at *E(spl)-C* while CSL enrichment remained (**Figure 5G and H**). During the subsequent activation and imaging, recruitment of Mam occurred more rapidly in preactivated cells in comparison to naïve cells, suggesting that the former are primed to rapidly reform an active hub containing Mam (**Figure 5I**).

Together, our data indicate that CSL recruitment and increased chromatin accessibility persist after Notch removal and after the loss of Mam-containing activation complexes. This persistent CSL confers a memory state that enables re-assembly of an activation hub and more rapid initiation of transcription in response to subsequent Notch activity.

## Probability of transcription conferred by Mastermind

When analysing the smFISH data, we noticed that, even in Notch ON conditions, a fraction of nuclei lacked foci of nascent transcription at *E(spl)-C*. Since Mam was present at *E(spl)-C* in all nuclei (**Figure 6A**, **Figure 6—figure supplement 1A**), this led us to question whether the presence of Mam complexes was sufficient to recruit downstream factors required for transcription initiation. We, therefore, investigated the extent that RNA Pol II was recruited to *E(spl)-C* in Notch ON cells using endogenous GFP::Rbp3 (**Cho et al., 2022**). When scanning all nuclei, it was evident that Rbp3 enrichment at *E(spl)-C* in Notch ON cells was highly variable. Robust enrichment was detected in a subset of nuclei, but, in other cases, there was little/no enrichment above nuclear levels (**Figure 6—figure supplement 1B**). Performing a Gaussian population fitting on the data, two populations gave the best fit and, based on these, 36% of nuclei had significant enrichment and 64% had similar levels to Notch OFF (**Figure 6B**, **Figure 6—figure supplement 1B**). This differs from Mam where all nuclei fall into a single population that has significant Mam enrichment at *E(spl)-C* (**Figure 6A**, **Figure 6—figure supplement 1A**). The striking difference in the proportions of nuclei with Pol II and with Mam enrichment implies

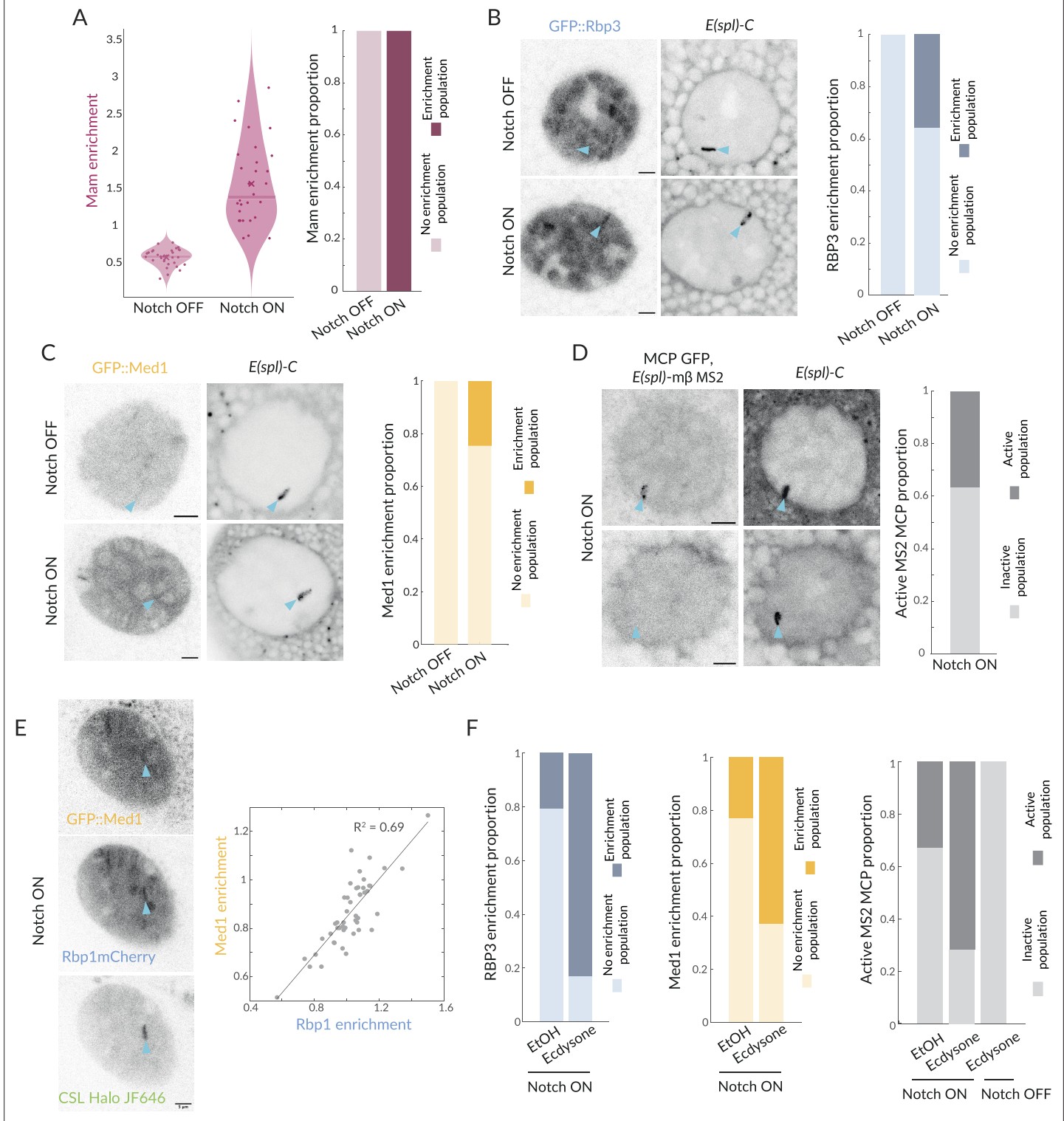

**Figure 6.** Recruitment of Pol II and Med1 in Notch ON is infrequent and augmented by ecdysone. (**A**) GFP::Mam enrichment at *E(spl)-C* in Notch OFF and Notch ON conditions as in *Figure 1*, represented with a violin plot with mean marked by a cross and median by a bar. Right graph: proportions of nuclei with significant enrichment (obtained by performing a Gaussian fit on the fluorescence intensity Images, see *Figure 6—figure supplement 1*). Over 90% nuclei have significant enrichment (Notch OFF, n = 32, Notch ON, n = 30). (**B**) Recruitment of GFP::Rbp3, a subunit of Pol II, at *E(spl)-C* in Notch OFF and Notch ON nuclei. Images: example where enrichment is detected. Graph: proportions of nuclei with significant enrichment as in (**A**) (Notch OFF, n = 37, Notch ON, n = 43). All scale bars represent 5 µm. (**C**) Recruitment of GFP::Med1 at *E(spl)-C* in Notch OFF and Notch ON nuclei. Images: example where enrichment is detected. Graph: proportions of nuclei with significant enrichment as in (**A**) (Notch OFF and ON, n = 38,

*Figure 6 continued on next page*

*Figure 6 continued*

62). All scale bars represent 5 µm. (**D**) Transcriptional activity in Notch ON cells detected using live imaging of GFP::MCP recruited to MS2 loops in transcripts produced by *E(spl)bHLH*mβ (see 'Materials and methods'). Images: representative examples of active (upper) and inactive (lower) nuclei. Graph: Proportion of active nuclei (Notch ON, n = 53). (**E**) Recruitment of GFP::Med1, mCherry::Rbp1, and Halo::CSL. Med1 and Rbp1 recruitments are correlated. Images: example of co-recruitment of Med1, Rbp1, and CSL. Graph: correlation plot of max enrichment per nucleus at *E(spl)-C*, (marked by Halo::CSL recruitment), n = 48. (**F**) Pre-treatment with ecdysone in Notch ON conditions increases proportions of nuclei with recruitment of GFP::Rbp3, GFP::Med1 and with active transcription foci (MS2/MCP intensity) compared to controls (EtOH). Proportions of nuclei with significant enrichment defined as in (**A**). Rbp3 ctrl, ecdysone: n = 39, 36. Med1 ctrl, ecdysone: n = 34, 34. MS2 ON ctrl, ON ecdysone, OFF ecdysone: n = 52, 42, 22.

The online version of this article includes the following figure supplement(s) for figure 6:

**Figure supplement 1.** Population fitting reveals infrequent enrichment of Pol II and Med1 but is augmented by ecdysone treatment.

that there is a limiting step, which results in transcription initiation being probabilistic/stochastic in these conditions.

It has been suggested that the core Mediator complex, as distinct from the CDK module, has an important role in bridging between enhancer-bound transcription complexes and initiation complexes at promoters (***Richter et al., 2022***; ***El Khattabi et al., 2019***; ***Soutourina, 2018***). To investigate core Mediator recruitment, we generated a GFP::Med1 fusion by CRISPR genome-editing. Using this endogenously tagged protein, which is homozygous viable, we found that GFP::Med1 became enriched at *E(spl)-C* with a similar probability to Rbp3 (***Figure 6C***). Indeed when we compared directly the enrichment of Pol II and Med1 in Notch ON nuclei, by using mCherry::Rbp1 (***Cho et al., 2022***) to monitor Pol II, we observed a significant correlation between the two proteins ($R^2$ = 0.69, ***Figure 6E***). No such correlation with CSL was observed, in the same experiments (***Figure 6—figure supplement 1E***).

To verify that only a subset of nuclei was transcriptionally active, we used a strain where MS2 loops have been inserted into *E(spl)*mβ using CRISPR-Cas9 engineering (***Boukhatmi et al., 2020***) and imaged transcription live in Notch ON nuclei. In the presence of MCP-GFP, which binds to MS2 loops in the RNAs produced, nascent sites of transcription appear as puncta that align into a band of fluorescence at *E(spl)-C* due to the multiple aligned gene copies. A robust band of MCP/MS2 nascent transcription at *E(spl)-C* was detected in Notch ON conditions, demonstrating that some nuclei were actively transcribing (***Figure 6D***, ***Figure 6—figure supplement 1D***). However, this was present in only a third (37%) of nuclei. These data show that only a subset of nuclei engage in active transcription, and that the proportion of active nuclei is similar to that with recruitment of Pol II and Med1.

These data demonstrate that the presence of Mam complexes is not sufficient to reliably drive all the steps required for transcription in every Notch ON nucleus. Instead, it appears that transcription is initiated stochastically. Based on previous study in the *Drosophila* embryo, where transcription in responding nuclei was highly synchronised (***Falo-Sanjuan et al., 2019***), this probabilistic outcome was unexpected although stochastic transcriptional responses have been observed in *C. elegans* (***Lee et al., 2019***). We note that such properties can only be detected using in vivo imaging approaches to monitor the responses of individual nuclei, as we have done here.

## Ecdysone cooperates with Notch to increase probability of transcription

Even though relatively few Notch ON nuclei became transcriptionally active in our experiments, they all had robust recruitment of Mam complexes and of Med13 at *E(spl)-C*. Thus, in many respects the gene locus becomes competent or poised for transcription in all nuclei. We wondered, therefore, whether the presence of a second stimulatory signal would increase the probability of loci becoming transcriptionally active. In normal development, salivary glands become exposed to the steroid hormone ecdysone a few hours after we perform our experiments. Two observations suggested that ecdysone was a good candidate for a cooperating signal. First, previous genome-wide studies detected ecdysone receptor binding in the *E(spl)-C* region (***Uyehara et al., 2022***). Second, we noticed in rare samples containing an older gland that it had a higher proportion of active nuclei. We, therefore, exposed the early-stage Notch ON salivary glands to ecdysone and analysed the proportion of nuclei with MS2/MCP puncta. Strikingly, the proportion of active nuclei increased dramatically to 70% following ecdysone treatment. No such effects were seen in Notch OFF nuclei where *E(spl)*mβ remained silent even after ecdysone treatment (***Figure 6F***, ***Figure 6—figure supplement 1H***).

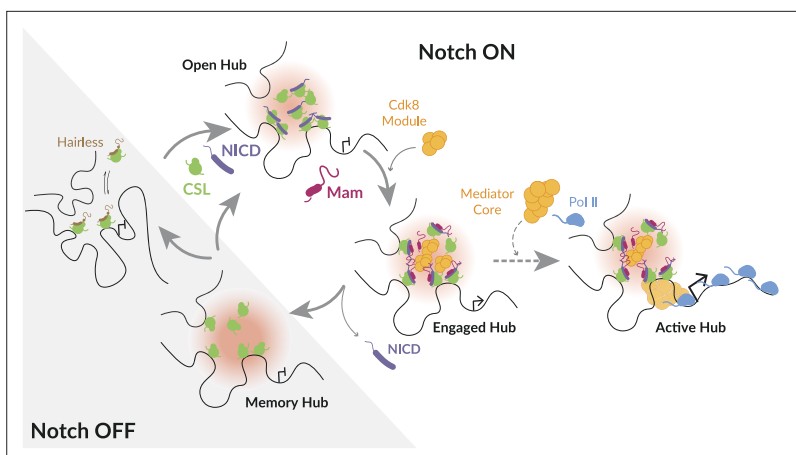

**Figure 7.** Model illustrating different modes of Notch transcription hub. In the absence of Notch activity (Notch OFF), target genes are inactive, CSL is complexed with co-repressors (e.g. Hairless). Following Notch activation (Notch ON), released Notch Intra Cellular Domain (NICD) (purple) generates a localised high concentration of transcription factors around target enhancer(s) referred to as a 'hub' (pink). Open hub: CSL (green) recruitment and accessible chromatin can occur in the absence of Mam. Engaged hub: presence of Mam (magenta) favours recruitment of additional factors, including Mediator CDK8 module (orange). Active hub: productive transcription, transition to this mode, with core Mediator (orange) and Pol II (pale blue) enrichment, is stochastic (dotted arrow). The probability can be enhanced by secondary signal, such as provided by ecdysone. Memory hub: CSL enrichment and chromatin accessibility remain after withdrawal of NICD.

We further tested the effect of ecdysone by measuring changes in enrichment of Pol II and Med1 at *E(spl)-C*. The proportion of nuclei with clear enrichment of each factor was significantly increased (*Figure 6F*, *Figure 6—figure supplement 1F and G*). Thus, the switch from the more stochastic transcription elicited by recruitment of Mam alone to the more robust initiation when ecdysone was added, correlated with the presence of Med1 and Pol II. As Mediator is reported to stabilise enhancer–promoter interactions, its recruitment may be what limits the probability of transcription (*Richter et al., 2022*; *Luyties and Taatjes, 2022*).

## Discussion

Understanding the mechanisms by which signalling levels and dynamics are accurately converted to transcriptional outputs is fundamental for developmental programming. Here we used live imaging approaches to probe how this occurs in the context of the Notch pathway, where the transcriptional relay relies on the single NICD released by each activated receptor and the nuclear complexes it forms with its partners CSL and Mam. By tracking these complexes in real time, we unveiled three important features (*Figure 7*): (i) in Notch ON nuclei, the activation complexes promote formation of a dynamic protein hub at a regulated gene locus that concentrates key factors including Mediator CDK module; (ii) the composition of the hub is changeable and the footprint persists after Notch withdrawal conferring a memory that enables rapid reactivation; and (iii) transcription is probabilistic in Notch ON nuclei, such that only a third of nuclei with a hub are actively transcribing. This has far-reaching implications because it reveals that stochastic differences in Notch pathway output can arise downstream of receptor activation.

### Notch transcription complexes form a hub

Many recent studies have demonstrated that hubs or condensates play key roles in gene-expression regulation by maintaining a high local concentration of transcription factors and other regulatory complexes (*Sabari et al., 2018*; *Demmerle et al., 2023*; *Boija et al., 2018*; *Richter et al., 2022*). By tracking the co-activator Mam in real time, we show that Notch transcription complexes become enriched around a target locus in a zone that has characteristics of a hub. First, the high-concentration zones are highly dynamic and undergo a continual exchange between activator and repressor complexes. The former exhibit longer residence times, suggesting that they are stabilised by other

interactions. Second, recruitment of CSL-containing complexes is non-stoichiometric with respect to CSL-binding motifs and can in part be recapitulated by an IDR from NICD. Although deletion of the IDR biases against activation complex recruitment, our evidence suggests that individual IDRs make a minor contribution, as consistent with an unbiased study of IDRs (*Hannon and Eisen, 2023*). Third, the enriched zone is readily detectable by live imaging, but is sensitive to fixation, as has been reported for some subcellular protein condensates (*Irgen-Gioro et al., 2022*). Although transcription hubs have frequently been associated with phase separation, examples are emerging where high local concentrations of transcription factors can be achieved without it (*Trojanowski et al., 2022*). Similarly, the locally enriched Notch-induced hubs in our experiments do not manifest clear properties of liquid condensates despite being non-stoichiometric.

We propose that NICD and Mam participate in multivalent interactions, which increase the likelihood of the tripartite activation complexes remaining in the chromatin-associated state and which facilitate recruitment of co-activators to form a local hub (*Figure 7*). The latter include subunits of CDK module, Med13. We find that perturbations to CDK module compromise Mam levels in the hub and vice versa, highlighting its importance. Indeed, the CDK module has been reported to interact directly with Mam and to play a role in Notch signalling output (*Fryer et al., 2004*; *Janody and Treisman, 2011*), and recent studies in mammalian cells indicate CDK8 acts positively in signal-induced gene expression (*Chen et al., 2023*). Although its recruitment is proposed to prepare genes for activation, the transition to activation necessitates CDK8 release since CDK module sterically inhibits core Mediator-Pol II interactions (*Luyties and Taatjes, 2022*; *Osman et al., 2021*). This is consistent with our results showing that Med13 recruitment is highly dynamic and that, despite most nuclei showing Med13 enrichment within the hub, the probability of transcription is low, as discussed below.

## Role of hub in conferring transcriptional memory

Assembly of the transcription hub at *E(spl)-C* depends on Notch activity but, surprisingly, can form in the absence of the co-activator Mam, albeit the composition differs. Notably, CSL complexes continue to be highly enriched and the chromatin at *E(spl)-C* enhancers is accessible in the absence of Mam (*Figure 7*). These properties also persist for many hours after withdrawal of Notch activity and in this way enhancers that have been switched on by Notch retain an imprint that can influence their response to a subsequent signal, a phenomenon known as transcriptional memory (*Bonifer and Cockerill, 2017*; *Avramova, 2015*). As an indication, after a previous Notch activation event, the conversion to an activation hub with Mam enrichment occurs more rapidly than in the absence of a prior active state and there is enhanced transcriptional activity. Similar accelerated recruitment of STAT1 to promoters occurred in cells with IFNγ-induced transcriptional memory (*Tehrani et al., 2023*), and augmented transcription levels have been seen in several contexts (*Ferraro et al., 2016*; *Zhao et al., 2020*).

The concept that enhancers retain a memory of a recent Notch signal has several implications. First, naïve and preactivated loci will respond with different kinetics, which could bias cell-fate decisions when iterative Notch signals are deployed (*Pourquié, 2003*). Second, although our analysis is focused on post-mitotic cells, the footprint may be inherited by daughter cells as CSL is retained on mitotic chromosomes in at least some cell types (*Dreval et al., 2022*). If this is the case, it may explain why enhancer decommissioning is important to switch off the Notch response in some contexts (*Zacharioudaki et al., 2019*; *Jacobs et al., 2022*; *Oldershaw et al., 2008*). Third, it could explain some of the observed recruitment of CSL co-repressor complexes to active chromatin if they are involved in sustaining the hub when the signal decays (*Oswald et al., 2016*; *Chan et al., 2017*).

## Transcription initiation is probabilistic

Because the in vivo live imaging enables us to monitor the responses of individual nuclei, we made the surprising discovery that the presence of Mam-containing hubs at *E(spl)-C* is not sufficient to guarantee transcription. Instead, our results reveal that Mam recruitment confers a 1 in 3 chance of transcription. As transcription initiation is inherently stochastic (*Lammers et al., 2020*; *Meeussen and Lenstra, 2024*), it is not necessarily unexpected that only a fraction of nuclei would be active at any one time. Indeed, stochastic response to Notch activity has been detected in the *C. elegans* gonad, where a similarly probabilistic response dependent on Notch was detected (*Lee et al., 2019*; *Lee et al., 2016*). However, this situation differs from the *Drosophila* mesectoderm, where the response to Notch was fully penetrant and highly synchronised across multiple nuclei, arguing that Notch-regulated

transcription is not a priori stochastic (*Falo-Sanjuan et al., 2019*). Furthermore, it has been suggested that recruitment of the activation complex would be the limiting factor. In our context it is evident that recruitment of Mam activation complexes is not the limiting step. This also differs from experiments with the glucocorticoid receptor, which concluded that its binding determined the frequency and length of the RNA bursts produced (*Stavreva et al., 2019*).

The observation that at any one time only a third of nuclei are actively transcribing could be a reflection either of two cell populations, in which only a subset of cells transition to the ON state or two dynamic states in which all nuclei transition between ON and OFF states. Monitoring transcription live for prolonged periods, we rarely detected transitions in the same nucleus, arguing that, if the differences reflect the dynamic states, both the ON and the OFF state must be of prolonged duration. At the same time, the live traces reveal that the transcription in the ON cells is dynamic, in keeping with frequent fluctuations in promoter states producing 'bursts' (*Lammers et al., 2020*; *Meeussen and Lenstra, 2024*). Thus we would argue that there are two probabilistic steps involved. The first is the transition of the target gene into an active ON state, which, in our basic Notch active condition, appears relatively infrequent, and the second is the switching ON and OFF of the promoter in ON cells to produce 'bursts' of RNA, which is much more rapid. The presence of a second signal, in the form of ecdysone, was sufficient to increase the probability of the first, slow, transition occurring.

In our experiments, the probability of nuclei being in the ON state correlated with the proportion of nuclei where Med1, a core Mediator subunit, was recruited into the hub at *E(spl)-C*. Mediator is an important intermediary between transcription factors and the preinitiation complex, and it functions as an 'integrator' coordinating diverse and combinatorial inputs (*Richter et al., 2022*). The increased probability of Mediator recruitment can thus be explained by an increase in the valency of possible interactions when Notch and ecdysone are both active and present at the enhancers of target genes. We propose that this will be a general mechanism and that other signals/transcription factors will synergise with Notch complexes by adding to the valency of interactions and facilitating recruitment of co-activators to increase the probability of transcription. In different contexts this could toggle the response from one that is stochastic to one that is hardwired.

The observation of a probabilistic transcriptional outcome downstream of Notch activity and Mam recruitment has profound implications because it has widely been assumed that stochastic differences in Notch pathway output arise due to fluctuations in ligands or ligand availability (*Nandagopal et al., 2018*; *Kageyama et al., 2008*). Our results raise the possibility that differences may also arise due to intrinsic variations subsequent to/independent from receptor activation that affect the probability of transcription occurring. Furthermore, this probability can be modified by other signals, as we observed here with ecdysone, or by the presence of other transcription factors, as when enhancers are 'primed' (*Falo-Sanjuan et al., 2019*) and/or cooperatively regulated (*Drier et al., 2016*; *Terriente-Felix et al., 2013*; *Wang et al., 2014*; *Larrivée et al., 2012*; *Mittal et al., 2009*).

## Materials and methods

**Key resources table**

| Reagent type (species) or resource | Designation | Source or reference | Identifiers | Additional information |
| --- | --- | --- | --- | --- |
| Genetic reagent (*Drosophila melanogaster*) | eGFP::CSL | *Gomez-Lamarca et al., 2018* | FBgn0004837 | Genomic fragment insertion attP86Fb (RRID:BDSC24749) |
| Genetic reagent (*D. melanogaster*) | Halo::CSL | *Townson et al., 2023* | FBgn0004837 | Genomic fragment insertion attP86Fb |
| Genetic reagent (*D. melanogaster*) | eGFP::Hairless | *Gomez-Lamarca et al., 2018* | FBgn0001169 | Genomic fragment insertion attP51D |
| Genetic reagent (*D. melanogaster*) | Hairless::Halo | *Baloul et al., 2024* | FBgn0001169 | Genomic fragment insertion attP51D |
| Genetic reagent (*D. melanogaster*) | sfGFP::Mam | This paper | FBgn0002643 | CRISPR of Mam locus |
| Genetic reagent (*D. melanogaster*) | Halo::Mam | This paper | FBgn0002643 | CRISPR of Mam locus |

*Continued on next page*

*Continued*

| Reagent type (species) or resource | Designation | Source or reference | Identifiers | Additional information |
|---|---|---|---|---|
| Genetic reagent (*D. melanogaster*) | Med13::YFP | Bloomington Drosophila Stock Center | RRID:BDSC_57899 | |
| Genetic reagent (*D. melanogaster*) | eGFP::Rbp3 | *Cho et al., 2022* | | CRISPR of Rbp3 locus |
| Genetic reagent (*D. melanogaster*) | mCherry::Rbp1 | *Cho et al., 2022* | | CRISPR of Rbp1 locus |
| Genetic reagent (*D. melanogaster*) | UAS-Hairless-RNAi | Bloomington Drosophila Stock Center | RRID:BDSC_27315 | Line used in: *Gomez-Lamarca et al., 2018* |
| Genetic reagent (*D. melanogaster*) | UAS-Mam-RNAi | Bloomington Drosophila Stock Center | RRID:BDSC_28046 | Line used in: *Gomez-Lamarca et al., 2018*; *Lobo-Pecellín et al., 2019*; *Jia et al., 2016* |
| Genetic reagent (*D. melanogaster*) | UAS-Med13-RNAi | Bloomington Drosophila Stock Center | RRID:BDSC_34630 | Line used in: *Ren et al., 2022* |
| Genetic reagent (*D. melanogaster*) | UAS-nejire-RNAi | Vienna Resource Center | | KK10288 |
| Genetic reagent (*D. melanogaster*) | UAS-Cdk8-RNAi | *Li et al., 2020* | | |
| Genetic reagent (*D. melanogaster*) | UAS-Mam[DN] | Bloomington Drosophila Stock Center *Helms et al., 1999* | | |
| Genetic reagent (*D. melanogaster*) | UAS-Cry2-TevC | Bloomington Drosophila Stock Center *Townson et al., 2023* | | Insertion attP51C |
| Genetic reagent (*D. melanogaster*) | UAS-OptIC-Notch{ω}[mCherry] | *Townson et al., 2023* | | Insertion attP40 |
| Genetic reagent (*D. melanogaster*) | *E(spl)-mβ*-HLH-MS2-LacZ | This paper, modified from *Boukhatmi et al., 2020* | FBgn0002733 | CRISPR of E(spl)-*mβ* |
| Chemical compound, drug | Triptolide | Sigma-Aldrich | T3652 | Used at 10 µM |
| Chemical compound, hormone | Ecdysone | Cayman Chemicals | 16145 | Used at 5 µM |
| Chemical compound, drug | Senexin A | Tocris | 4875 | Used at 1 µM |
| Chemical compound, drug | Senexin B | Cayman Chemicals | 24119 | Used at 2 µM |
| Commercial assay or kit | Tagmentation kit | Illumina | FC-121-1030 | |
| Software, algorithm | MATLAB | MathWorks | MATLAB R2022b | |

## Experimental animals

Species: *Drosophila melanogaster*. Flies were grown and maintained on food consisting of the following ingredients: glucose 76 g/l, cornmeal flour 69 g/l, yeast 15 g/l, agar 4.5 g/l, and methylparaben 2.5 ml/l. Animals of both sexes were used in this study.

## Fly stocks

For genetic manipulations of Notch activity, the Gal4 driver line *1151-Gal4* was combined with *UAS-NΔECD* to provide constitutively active Notch (*Rebay et al., 1993*; *Fortini et al., 1993*) or with *UAS-LacZ* as a control. In experiments with fluorescent labeled proteins the following were used: GFP::CSL and GFP::Hairless (*Gomez-Lamarca et al., 2018*), Halo::CSL, Med13::YFP (BL-57899). Experiments measuring in vivo recruitment to *E(spl)-C* utilised a 'locus tag' chromosome in which Int1 sequences had been inserted into an *E(spl)-C* intergenic region and recombined with *UAS-ParB1-mcherry* or *UAS-ParB1-GFP* inserted AttP.86Fb (*Gomez-Lamarca et al., 2018*).

To manipulate protein functions, RNAi lines were as listed in *Supplementary file 1—Table 2* and included *UAS-Hairless-RNAi* (Bloomington *Drosophila* Stock Center, BL-27315), *UAS-Mam-RNAi*

(BL-28046), *UAS-Med13-RNAi* (BL-34630), *UAS-nejire-RNAi* (Vienna *Drosophila* Resource Center, VDRC-102885), *UAS-Cdk8-RNAi* (gifted by *Li et al., 2020*) or with *UAS-MamDN* to block Mam activity (*Helms et al., 1999*). Controls as appropriate for each chromosome were *UAS-yellow-RNAi* (II), *UAS-white-RNAi* (III). Crosses were maintained at 25°C and knock-down was validated by RT-qPCR (*Figure 1—figure supplement 1D* and *Figure 3—figure supplement 1J*).

For temperature manipulations, *MS1096-Gal4* (BL-8860) was recombined with *tubulin-Gal80^{ts}* and flies were switched between 29°C non-permissive (Gal80^{ts} inactive, Notch ON) and 18°C permissive (Gal80^{ts} active, Notch OFF) temperatures.

For photomanipulation, transgenes *UAS-Cry2-TevC* and *UAS-OptICNotch{ω}* were recombined onto one chromosome and the conditions used were as described (*Townson et al., 2023*) and see below.

For MCP/MS2 live imaging of transcription, a strain in which 24M2 loops were inserted into *E(spl)* mβ-HLH gene was generated by CRISPR using the strategy described (*Boukhatmi et al., 2020*); details are provided in *Supplementary file 1—Table 1*. This was combined with *hsp83-MCP::GFP* (BL-7280).

### Generation of tagged Mastermind and Med1 flies

CRISPR/cas9 genome engineering was used to introduce fluorescent (sfGFP) or Halo tags into N-terminal coding regions of Mam and Med1 (flycrispr.org) to generate seamless protein fusions. Briefly, plasmids for expression of the gRNA (pCFD3-dU6) and for homology arm repair (pHD-ScarlessDsRED) were injected into *nanos-cas9* flies (flycrispr.org). Transformants were selected based on the expression of 3xPax3-dsRED which was subsequently removed by crossing to *αTub84B-PiggyBac* flies (BL-32070). Details of gRNA and homology sequences are provided in *Supplementary file 1—Table 1*.

### Generation of flies expressing IDR-GFP fusions

IDR regions of CSL, NICD, Med1, and Mam were isolated from genomic DNA by PCR (*Supplementary file 1—Table 1*) and inserted into the plasmid *pUASt-attB* (DGRC, 1419). sfGFP was inserted into the N terminus so that the coding sequences generated an in-frame protein fusion. The resulting plasmids contained an attB sequence and were injected into a strain containing phiC31 integrase and AttP site in position *AttP40* (chromosome II, BL-25709) to obtain transgenic flies for conditional expression of the IDR fusions.

### Method details

#### Salivary gland culture and drug/hormonal treatments

Salivary glands were dissected and mounted as described (*Gomez-Lamarca et al., 2018*) by submerging mid L3 larvae in M3 Shields and Sang media (Sigma-Aldrich, S3652) supplemented with 5% fetal bovine serum (Sigma-Aldrich, F9665) and 1× Antibiotic-Antimycotic (Gibco, 15240-062). For drug and hormonal treatments, dissected glands were incubated for an hour with the following compounds: triptolide (10 μM, Sigma-Aldrich T3652), ecdysone (5 μM, Cayman Chemicals 16145), A485 (5 μM, Cayman Chemicals 24119), Senexin A (1 μM, Tocris 4875), and Senexin B (2 μM, Cayman Chemicals 24119). After dissection and any treatments glands were mounted into polylysine-coated coverslips in dissection media supplemented with methyl-cellulose (Sigma-Aldrich, M0387-100G).

#### Optogenetic Notch activation

The OptIC-Notch{ω} system was used (*Townson et al., 2023*) in which blue light conditions induce association of two components, CRY2-TEVc and CIBN-TEVn-mCherry-NICD, which leads to the cleavage and release of NICD, mimicking Notch activation. Larvae were maintained under strict dark conditions and for imaging were dissected under an amber light to maintain OFF conditions. Activation was achieved by transferring larvae to a blue light incubator for the indicated times or by exposure to a blue laser (458 nm) during live imaging, as described previously (*Townson et al., 2023*).

#### Confocal imaging and FRAP analysis

Live confocal fluorescence imaging of salivary glands was performed on a Leica SP8 microscope equipped with seven laser lines (405, 458, 488, 496, 514, 561, and 633) and a ×63/1.4 NA HC PL APO CS2 oil immersion objective and two hybrid GaAsP detectors. Individual nuclei were imaged with a 4.5× zoom, 512 × 512 pixel resolution, pinhole set to 3-Airy, three line averages, a 12-bit depth, and

600 Hz scanning speed. Z-stacks were chosen to encompass the locus and around nine stacks were acquired. The step size was chosen based on pinhole aperture and was <1 um. FRAP was performed on the same microscope but settings were optimised for bleaching and scanning speed: pinhole was opened to 3.5-Airy, speed was increased to 700 Hz, line averaging was removed, and Leica FRAP booster was activated. Effective bleaching was achieved by point bleaching. Images before and after bleaching were acquired every 0.4 s. After 50 images post bleaching, frame gap was increased to 1 s to minimise unintentional bleaching.

FRAP curves were normalised as described (*Gomez-Lamarca et al., 2018*) by applying the following:

$$Recovery = \frac{T_{pre} \times B_t}{T_t \times B_{pre}}$$

where intensities are *Tpre* in a nuclear region before bleaching, *Bt* in the bleached region throughout the experiment, *Tt* in a nuclear region throughout the experiment, and *Bpre* the bleached region throughout the experiment.

## Single-particle imaging

Sample preparation and imaging for SPT experiments were performed as in *Baloul et al., 2024*. Briefly after dissection, glands were incubated with Halo ligand, TMR (Promega, G825A) for 15 min and washed in three consecutive 10 min baths of dissecting medium. Halo ligand concentrations used were 10 nM, 10 nM, 50 nM, and 0.01–0.02 nM for CSL, Hairless, Mastermind, and Histone H2AV, respectively. Larvae were imaged on custom-build inverted microscope optimised for Single-Molecule Localisation Microscopy (*Gomez-Lamarca et al., 2018*; *Baloul et al., 2024*) with 50 ms exposure time for 3–7 min approximately per nuclei.

## Immunofluorescence and in situ hybridisation

Salivary glands were prepared for immunofluorescence as described previously (*Gomez-Lamarca et al., 2018*). Larvae were submerged in PBS and salivary glands were dissected and then fixed in 4% formaldehyde for 15 min. Glands were washed three times in PBS +0.3% Triton X-100. They were later blocked by adding 1% BSA to this buffer. Primary antibody incubation was performed overnight at 4°C with αGFP (1/500, Thermo Fisher A6455) and αRFP (1/1000, Chromotek 5F8), αH3K27ac (1/500, ActiveMotif 39135). Glands were washed at least three times with BSA containing buffer, followed by secondary antibody and nuclear stain incubation for 2 hr at room temperature (RT) (Jackson Immu-noResearch Laboratories, Inc and Sigma). Lastly, they were washed three times and mounted in Vecta-shield. Image acquisition was performed similarly to live imaging but with a with pinhole closed to 1 Airy and an optimised Z step size, a 0.75 zoom. and a 1024 × 1024 px resolution.

smFISH probes were designed with Stellaris Probe tool for *E(spl)m3-HLH* and salivary glands were processed as described (*Boukhatmi et al., 2020*). Briefly, glands were fixed for 45 min in 3.7% form-aldehyde at RT and permeabilised overnight in 70% EtOH at 4°C. Hybridisation and washes were performed according to the manufacturer's instructions. Image acquisition was performed similarly to live imaging but with a with pinhole closed to 1 Airy and an optimised Z step size and a 10× zoom. Cytoplasmic images always contained constant Z steps.

## ATAC qPCR

Accessibility of genomic regions was probed by tagmentation reaction coupled with qPCR, as described (*Gomez-Lamarca et al., 2018*). Briefly, salivary glands were lysed and nuclei were suspended in TD buffer and TD DNA tagment enzyme was added (Illumina #FC-121-1030). The chro-matin was tagmented for 30 min at 37°C. DNA was amplified with Nextera primers, and samples were normalised by running qPCRs and determining the necessary extra cycles for each sample. Standard qPCRs were performed to quantify accessibility of different regions (primers indicated in *Supplementary file 1—Table 1*, Roche #04707516001).

## RT qPCR

mRNA abundance was measured by retro-transcription coupled with qPCR as described (*Gomez-Lamarca et al., 2018*). Briefly, RNA was extracted with Tri Reagent (Invitrogen #AM9738), precipitated,

and DNAse treated (Invitrogen #AM1906). cDNA was generated according to the manufacturer's instructions with M-MLV Reverse Transcriptase (Promega #M1701). Standard qPCRs were performed to measure RNA levels (Roche #04707516001).

## Analysis

### Confocal image analysis

Images were analysed using MATLAB by importing Leica Images with BioFormat package (MATLAB R2022b, MathWorks and openmicroscopy.org). A custom MATLAB app was built to select and rotate the Z stack of interest. A rectangular region of interest of 90 × 40 pixels was selected for each nucleus, centred on the ParA/B recruitment to IntA/B site in *E(spl)-C* identifiable in one of the channels. Additionally, three circular regions were drawn to measure the nuclear levels in the selected stack, which by division or subtraction enabled normalisation of the rectangular region of interest (ROI). The ROIs obtained are centred, and this allowed averaging experimental conditions, referred to as 'averaged intensities or enrichment'. For the profiles, the ROIs were averaged in the y-dimension, and the mean and SEM were represented for each condition. Lastly, for the max enrichment the 10 middle pixels were obtained from the intensity or enrichment profile. In all figures, the position of *E(spl)-C* is indicated by a grey bar above average intensity images and by grey shading/0 in enrichment profile plots.

To obtain proportions of enrichment cells, Gaussian fitting of enrichment values was applied, where the proportion, mean, and sigma were used to characterise the enrichment populations. In the correlation, $R^2$ is the proportion of variance of y explained by the variance of x calculated as 1-RSS/TSS, where RSS = sum of squared residuals and TSS = total sum of squares.

### SPT analysis

SMLM movies were analysed using the pipeline described in *Baloul et al., 2024*. Localisation of single molecules was carried out using a Gaussian fitting-based approach (*Ovesný et al., 2014*) while a multiple hypothesis-tracking algorithm based on *Chenouard et al., 2013* was used for tracking. No detection gaps were allowed in tracking except in the case of analyses focusing solely on the duration of trajectories (*Figure 1E and E'*), for which up to three detection gaps were allowed. Trajectories consisting of at least four time points were then analysed with a Bayesian treatment of HMM, vbSPT, and assigned into two states, each defined by a Brownian motion diffusion coefficient.

Trajectory density analysis shown in *Figure 1D and E'* was carried out by comparing near-locus density with nuclear density using the formula

$$\frac{No. of trajectories near locus}{Total no. of trajectories \in nucleus} \times \frac{Nucleus area}{Locus area}.$$

Locus and nucleus areas were calculated with standard MATLAB procedures using the convex hull of localisations and masking.

1-CDF survival curves (*Figure 1E and E'*) were plotted using 99% of trajectories for each molecule, excluding the longest 1% of trajectories which could be artefacts and bias the data.

## Statistical analysis

N numbers indicate the number of nuclei images, unless stated otherwise. If n > 30 for both conditions tested, a two-tailed *t*-test was applied. Otherwise, normality was checked via a Shapiro–Wilk test. If both samples were not normal, a Wilcoxon rank sum test was applied. In all cases, significance was presented as follows: *p<0.05, **p<0.01, ***p<0.001, and ****p<0.0001, and p-values are provided in *Supplementary file 1—Table 4*.

## Acknowledgements

We thank Kat Millen and the Genetics Department Fly Facility for performing DNA injections of fly embryos to generate transgenic stocks. We are grateful to Cambridge Advanced Imaging Centre for advice on imaging and to all members of the Bray Lab for helpful discussions and comments on the manuscript. Stocks obtained from the Bloomington Drosophila Stock Center (NIH P40OD018537) were used in this study. The work was funded by a Wellcome Trust Investigator Award (212207/Z/18) and an MRC Programme Grant (MR/T014156/1) to SJB. CR and SB, were supported by studentships

from Wolfson College-University of Cambridge (Dept of Physiology Development and Neuroscience-School of Biological Sciences).

## Additional information

### Funding

| Funder | Grant reference number | Author |
|---|---|---|
| Wellcome Trust | 10.35802/212207 | Sarah Bray |
| Medical Research Council | MR/T014156/1 | Sarah Bray |

The funders had no role in study design, data collection and interpretation, or the decision to submit the work for publication. For the purpose of Open Access, the authors have applied a CC BY public copyright license to any Author Accepted Manuscript version arising from this submission.

### Author contributions

F Javier DeHaro-Arbona, Conceptualization, Resources, Formal analysis, Validation, Investigation, Visualization, Methodology, Writing - original draft, Writing – review and editing; Charalambos Roussos, Conceptualization, Resources, Software, Formal analysis, Validation, Methodology, Writing - original draft, Writing – review and editing; Sarah Baloul, Conceptualization, Formal analysis, Validation, Investigation, Methodology, Writing - original draft, Writing – review and editing; Jonathan Townson, Conceptualization, Resources, Investigation, Methodology, Writing – review and editing; María J Gómez Lamarca, Conceptualization, Resources, Supervision, Methodology, Writing – review and editing; Sarah Bray, Conceptualization, Supervision, Funding acquisition, Visualization, Writing - original draft, Project administration, Writing – review and editing

### Author ORCIDs

F Javier DeHaro-Arbona  http://orcid.org/0000-0002-2307-0859
María J Gómez Lamarca  https://orcid.org/0000-0003-3750-7959
Sarah Bray  http://orcid.org/0000-0002-1642-599X

Reviewer #2 (Public Review): https://doi.org/10.7554/eLife.92083.3.sa1
Reviewer #3 (Public Review): https://doi.org/10.7554/eLife.92083.3.sa2
Author response https://doi.org/10.7554/eLife.92083.3.sa3

## Additional files

### Supplementary files

• Transparent reporting form

• Supplementary file 1. Genomic coordinates and oligonucleotides used for CRISPR, constructs, and qPCR (Table 1). Summary of *Drosophila* strains (Table 2). Genetic combinations for each figure (Table 3). p-values from statistical tests (Table 4).

### Data availability

Data generated and analysed during this study are included in the manuscript and the source raw images are available via FigShare.Code generated for image analysis is available on GitHub:https://github.com/BrayLab/sgv2/ (copy archived at *BrayLab, 2024*).The image data that supports these findings have been organised according to figures and deposited in an openly available FigShare project:https://figshare.com/projects/Dynamic_modes_of_Notch_transcription_hubs_conferring_memory_and_stochastic_activation_revealed_by_live_imaging_the_co-activator_Mastermind/197851. Previously generated Single Particle Analysis data was reanalysed (*Baloul et al., 2024*), already available in FigShare. https://figshare.com/projects/Changes_in_searching_behaviour_of_CSL_transcription_complexes_in_Notch_active_conditions/187665 and analysed with code available

in GitLab:https://gitlab.developers.cam.ac.uk/cr607/smt_trajectory_analysis. The materials generated in this study, which are new *Drosophila* strains, are available upon request.

The following dataset was generated:

| Author(s) | Year | Dataset title | Dataset URL | Database and Identifier |
|---|---|---|---|---|
| DeHaro Arbona FJ, Bray S | 2024 | Dynamic modes of Notch transcription hubs conferring memory and stochastic activation revealed by live imaging the co-activator Mastermind | https://figshare.com/projects/Dynamic_modes_of_Notch_transcription_hubs_conferring_memory_and_stochastic_activation_revealed_by_live_imaging_the_co-activator_Mastermind/197851 | FigShare, Dynamic_modes_of_Notch_transcription_hubs_conferring_memory_and_stochastic_activation_revealed_by_live_imaging_the_co-activator_Mastermind/197851 |

The following previously published dataset was used:

| Author(s) | Year | Dataset title | Dataset URL | Database and Identifier |
|---|---|---|---|---|
| Baloul S | 2023 | Changes in searching behaviour of CSL transcription complexes in Notch active conditions | https://figshare.com/projects/Changes_in_searching_behaviour_of_CSL_transcription_complexes_in_Notch_active_conditions/187665 | FigShare, Changes_in_searching_behaviour_of_CSL_transcription_complexes_in_Notch_active_conditions/187665 |

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
