## [Editor Report · eLife assessment]

This **fundamental** study advances our understanding of how Notch signalling activates transcription by analysing the dynamics of the Mastermind transcriptional co-activator and its role in the activation complex. The evidence is **compelling** and based on state-of-the-art methods with precise quantitative measurements.

---

## [Referee Report · Reviewer #2 (Public Review)]

The manuscript from deHaro-Arbona et al, entitled "Dynamic modes of Notch transcription hubs conferring memory and stochastic activation revealed by live imaging the co-activator Mastermind", uses single molecule microscopy imaging in live tissues to understand the dynamics and molecular determinants of transcription factor recruitment to the E(spl)-C locus in *Drosophila* salivary gland cells under Notch-ON and -OFF conditions. Previous studies have identified the major players that are involved in transcription regulation in the Notch pathway, as well as the importance of general transcriptional coregulators, such as CBP/P300 and the Mediator CDK module, but the detailed steps and dynamics involved in these processes are poorly defined. The authors present a wealth of single molecule data that provides significant insights into Notch pathway activation, including:

(1) Activation complexes, containing CSL and Mam, have slower dynamics than the repressor complexes, containing CSL and Hairless.

(2) Contribution of CSL, NICD, and Mam IDRs to recruitment.

(3) CSL-Mam slow-diffusing complexes are recruited and form a hub of high protein concentrations around the target locus in Notch-ON conditions.

(4) Mam recruitment is not dependent on transcription initiation or RNA production.

(5) CBP/P300 or its associated HAT activity is not required for Mam recruitment

(6) Mediator CDK module and CDK8 activity is required for Mam recruitment, and vice-versa, but not CSL recruitment.

(7) Mam is not required for chromatin accessibility but is dependent on CSL and NICD.

(8) CSL recruitment and increased chromatin accessibility persist after NICD removal and loss of Mam, which confers a memory state that enables rapid re-activation in response to subsequent Notch activation

(9) Differences in the proportions of nuclei with both Pol II and with Mam enrichment, which results in transcription being probabilistic/stochastic. These data demonstrate that presence of Mam-complexes is not sufficient to drive all the steps required for transcription in every Notch-ON nucleus.

(10) The switch from more stochastic to robust transcription initiation was elicited when ecdysone was added.

Overall, the manuscript is well written, concise, and clear, and makes significant contributions to the Notch field, which are also important for a general understanding of transcription factor regulation and behavior in the nucleus. The authors have satisfactorily addressed all my criticisms of their initial submission and therefore congratulate the authors on an excellent paper.

---

## [Referee Report · Reviewer #3 (Public Review)]

Summary:

DeHaro-Arbona and colleagues investigate the in vivo dynamics of Notch-dependent transcriptional activation with a focus on the role of the Mastermind (MAM) transcriptional co-activator. They use GFP and HALO-tagged versions of the CSL DNA-binding protein and MAM to visualize the complex, and Int/ParB to visualize the site of Notch-dependent *E(spl)-C* transcription. They make several conclusions. First, MAM accumulates at E(Spl)-C when Notch signaling is active, just like CSL. Second, MAM recruits the CDK module of Mediator but does not initiate chromatin accessibility. Third, after signaling is turned off, MAM leaves the site quickly but CSL and chromatin accessibility are retained. Fourth, RNA pol II recruitment, Mediator recruitment and active transcription were similar and stochastic. Fifth, ecdysone enhance the probability of transcriptional initiation.

Strengths:

The conclusions are well supported by multiple lines of extensive data that is carefully executed and controlled. A major strength is the strategic combination of *Drosophila* genetics, imaging and quantitative analyses to conduct compelling and easily interpretable experiments. A second major strength is the focus on MAM to gain insights into dynamics of transcriptional activation specifically.

Weaknesses:

Weaknesses were minor. and have been addressed in the revised manuscript.

---

## [Author Response]

The following is the authors’ response to the original reviews.

**Public Reviews:**

**Reviewer #1 (Public Review):**
In this manuscript by DeHaro-Arbona et al., the authors wish to understand how a signaling pathway (Notch) is dynamically decoded to elicit a specific transcriptional output. In particular, they investigate the kinetic properties of Notch-responsive nuclear complexes (the DNA binding factor CSL and its co-activator Mastermind (mam) along with several candidate interacting partners). Their experimental model is the polytene chromosome of the *Drosophila* salivary gland, in which the naturally inactive Notch can be artificially induced through the expression of a constitutively active form of Notch.The authors develop a series of CRISPR and transgenic lines enabling the live imaging of these complexes at a specific locus and in various backgrounds (genetic perturbations/drug treatments). This quantitative live imaging data suggests that Notch nuclear complexes form hubs, and the authors characterize their binding dynamics. Interestingly, they elegantly demonstrate that the content of these hubs and their kinetic properties can evolve, even within Notch ON cells. Hence, they propose the existence of distinct hubs, distinguishing an open (CSL), engaged (CSK-Mam), or active (CSL-Mam-Med-PolII) configuration in Notch ON cells and an inactive hub (in Notch OFF having previously been exposed to Notch) state, that would explain the surprising transcriptional memory that the authors observe hours after Notch withdrawal.

We thank the reviewer for this constructive summary of our work

**Reviewer #2 (Public Review):**
The manuscript from deHaro-Arbona et al, entitled "Dynamic modes of Notch transcription hubs conferring memory and stochastic activation revealed by live imaging the co-activator Mastermind", uses single molecule microscopy imaging in live tissues to understand the dynamics and molecular determinants of transcription factor recruitment to the E(spl)-C locus in *Drosophila* salivary gland cells under Notch-ON and -OFF conditions. Previous studies have identified the major players that are involved in transcription regulation in the Notch pathway, as well as the importance of general transcriptional coregulators, such as CBP/P300 and the Mediator CDK module, but the detailed steps and dynamics involved in these processes are poorly defined. The authors present a wealth of single molecule data that provides significant insights into Notch pathway activation, including:(1) Activation complexes, containing CSL and Mam, have slower dynamics than the repressor complexes, containing CSL and Hairless.(2) Contribution of CSL, NICD, and Mam IDRs to recruitment.(3) CSL-Mam slow-diffusing complexes are recruited and form a hub of high protein concentrations around the target locus in Notch-ON conditions.(4) Mam recruitment is not dependent on transcription initiation or RNA production.(5) CBP/P300 or its associated HAT activity is not required for Mam recruitment.(6) Mediator CDK module and CDK8 activity are required for Mam recruitment, and vice-versa, but not CSL recruitment.(7) Mam is not required for chromatin accessibility but is dependent on CSL and NICD.(8) CSL recruitment and increased chromatin accessibility persist after NICD removal and loss of Mam, which confers a memory state that enables rapid re-activation in response to subsequent Notch activation.(9) Differences in the proportions of nuclei with both Pol II and with Mam enrichment, which results in transcription being probabilistic/stochastic. These data demonstrate that the presence of Mamcomplexes is not sufficient to drive all the steps required for transcription in every Notch-ON nucleus.(10) The switch from more stochastic to robust transcription initiation was elicited when ecdysone was added.Overall, the manuscript is well written, concise, and clear, and makes significant contributions to the Notch field, which are also important for a general understanding of transcription factor regulation and behavior in the nucleus. I recommend that the authors address my relatively minor criticisms detailed below.

We thank the reviewer for their thorough and constructive summary of our work. We are glad that they overall found it insightful and interesting. Below we have addressed the points they have raised.

Page 7, bottom. The authors speculate, "It is possible therefore that, once recruited, Mam can be retained at target loci independently of CSL by interactions with other factors so that it resides for longer." Is it possible that another interpretation of that data is that Mam is a limiting factor?

As indicated our comment is a speculation and is based on the observations summarized in the paragraph. We are not entirely sure what the reviewer is proposing as an alternate model. However, if it relates to the relative concentrations of the different factors, this would not account for the differences in trajectory durations. And for most aspects of our analysis, K[off] has the most profound influence on the results. Furthermore, differences persist even when CSL levels are considerably reduced (as in conditions with Hairless RNAi).

Page 9. The authors write, "A very low level of enrichment was evident for... for the CSL Cterminus..". The recruitment of CSL ct IDR does not appear to be statistically significant or there is no apparent difference (Figure S2C), suggesting the CSL ct IDR does not play a role in enrichment.

We agree with the comments of the reviewer and have adjusted the text on page 9 accordingly.

Page 9. The authors write, "Notably, MamnIDR::GFP fusion was present in droplets, suggesting it can self-associate when present in a high local concentration (Figure S2B)." Is this result only valid for Mam nIDR or does full-length Mam also localize into droplets, as has been previously observed for full-length mammalian Maml1 in transfected cells?

We agree that the observed foci of MamL1 that have been detected in mammalian cells are interesting. We have not tried to replicate those data because the large size of Mam has made it challenging to produce a full-length form in over-expression. We note however that another portion of Mam, MamIDR, does not make droplets when over-expressed despite it containing a large section of the disordered region of the *Drosophila* Mam. We have now included a comment about the mammalian data in the text (page 9) to put our findings in context.

Previous studies in mammalian cells suggest that Maml1 is a high-confidence target for phosphorylation by CDK8, see Poss et al 2016 Cell Reports https://doi.org/10.1016/j.celrep.2016.03.030. By sequence comparison, does fly Mam have similar potential phosphorylation sites, and might these be critical for Mam/CDK module recruitment?

We thank the reviewer for highlighting this point. Indeed, we were very excited when we learnt that MamL1 was found to be a high confidence CDK8 target and we looked hard in the Mam sequence for potential phosphorylation sites. Sadly, there is very little conservation between the fly and the mammalian proteins beyond the helical region that contacts CSL and NICD. Furthermore, there are no identifiable putative CDK8 phosphorylation sites based on conventional motifs. It therefore remains to be established whether or not Mam is a direct target of the CDK8 kinase activity. We have added an explanatory comment in the text (page 11).

Page 11: The authors write, "The differences in the effects on Mam and CSL imply that the CDK module is specifically involved in retaining Mam in the hub, and that in its absence other CSL complexes "win-out", either because the altered conditions favour them and/or because they are the more abundant." Are the "other" complexes the authors are referring to Hairless-containing complexes? With the reagents the authors have in hand couldn't this be explicitly shown for CSLcomplexes rather than speculated upon?

The reviewer is correct that CSL complexes containing Hairless are good candidates to be recruited in these conditions. We have compared the levels of Hairless at E(spl)-C following treatments with Senexin and have not detected a difference. However, it appears that the high proportion of unbound Hairless makes it difficult to detect/quantify the enrichment at E(spl)-C. We have therefore taken a different strategy, which is to measure the recruitment of a mutant form of CSL that is compromised for Hairless binding. Recruitment of the mutant CSL is detected in Notch-ON conditions, but is significantly reduced/absent following Senexin treatment. These data favour the model proposed by the reviewer that in the absence of CDK8 activity, the CSL-Hairless complexes win out. These new data have been added in new Supplementary Figure S3F and S3G (and see text page 11)

Page 12/13: The authors write, "Based on these results we propose that, after Notch activity decays, the locus remains accessible because when Mam-containing complexes are lost they are replaced by other CSL complexes (e.g. co-repressor complexes)." Again, why not actually test this hypothesis rather than speculate? The dynamics of Hairless complexes following the removal of Notch would be very interesting and build upon previously published results from the Bray lab.

We thank the reviewer for this comment and we agree it’s possible that the proportion of Hairless complexes increases after Notch withdrawal. However, for the reasons outlined above, it is difficult to quantify changes in Hairless, (and our preliminary experiment did not reveal any large-scale effect) and because of the complexity of the genetics we cannot straightforwardly extend the experiment to analyze the behaviour of the mutant CSL as above. Therefore, at present, we cannot say whether the loss of Mam is compensated by an increase in Hairless. We hope in future to investigate the characteristics of the memory in more depth.

Page 13: The authors write, "As Notch removal leads to a loss of Mam, but not CSL, from the hub, it should recapitulate the effects of MamDN." While the data in Figure 5B seem to support this hypothesis, it's not clear to me that the loss of Mam and MamDN should phenocopy each other, bc in the case of MamDN, NICD would still be present.

We apologise that this sentence was a bit misleading. We have now rewritten it to improve accuracy (page 13) “As Notch removal leads to a loss of Mam, but not CSL, from the hub, we hypothesised it would recapitulate the effects of MamDN on chromatin accessibility and transcription of targets.”

The temporal dynamics for Mam recruitment using the temperature- and optogenetic-paradigms are quite different. For example, in the optogenetic time course experiments, the preactivated cells are in the dark for 4 hours, while in the temperature-controlled experiments, there is still considerable enrichment of Mam at 4 hours. For the preactivated optogenetic experiments, how sure are the authors that Mam is completely gone from the locus, and alternatively, can the optogenetic experimental results be replicated in the temperature-controlled assays? My concern is whether the putative "memory" observation is just due to incomplete Mam removal from the previous activation event.

We appreciate the concerns of the reviewer. However, we are confident that the 4-hour optogenetic inactivation is much more effective than the equivalent time for temperature shifts. The temperature sensitive experiment involves a longer decay, because not only the protein but also the mRNA has to decay to fully remove NICD activity. The optogenetic experiments, involve only protein decay and so are more acute. Furthermore, we have tested (and we show in Figure 5H) that Mam is fully depleted after 4 hours “Off” in the optogenetic experiments.

In order to further strengthen the evidence in favour of the memory hub, we have extended the time-frame further to show that CSL is retained at the locus even after 24 hours “Notch OFF” in both the temperature and the optogenetic paradigm. We have also measured the effects on transcription after a 24hr OFF period using the optogenetic paradigm and seen that robust transcription is initiated in cells that have experienced a previous activation (preactivated) compared to those that have not (naïve). These new data have been added to new Figure 5 C-F and strongly support the memory model.

**Reviewer #3 (Public Review):**
Summary:DeHaro-Arbona and colleagues investigate the in vivo dynamics of Notch-dependent transcriptional activation with a focus on the role of the Mastermind (MAM) transcriptional co-activator. They use GFP and HALO-tagged versions of the CSL DNA-binding protein and MAM to visualize the complex, and Int/ParB to visualize the site of Notch-dependent E(Spl)-C transcription. They make several conclusions. First, MAM accumulates at E(Spl)-C when Notch signaling is active, just like CSL. Second, MAM recruits the CDK module of Mediator but does not initiate chromatin accessibility. Third, after signaling is turned off, MAM leaves the site quickly but CSL and chromatin accessibility are retained. Fourth, RNA pol II recruitment, Mediator recruitment, and active transcription were similar and stochastic. Fifth, ecdysone enhances the probability of transcriptional initiation.Strengths:The conclusions are well supported by multiple lines of extensive data that are carefully executed and controlled. A major strength is the strategic combination of *Drosophila* genetics, imaging, and quantitative analyses to conduct compelling and easily interpretable experiments. A second major strength is the focus on MAM to gain insights into the dynamics of transcriptional activation specifically.

We thank the reviewer for their positive comments about the strengths of our work.

Weaknesses:Weaknesses are minor. There were no p-values reported for data presented in Figure S1D and no indication of how variable measurements were. In addition, the discussion of stochasticity was not integrated optimally with relevant literature.

We thank the reviewer for noting these points. The statistical tests have now been included for Figure S1D (now Figure S1F). We have amplified the discussion about stochasticity, to include more reference to the literature and to make clear also the distinction with transcription bursting (page 19, 20).

**Recommendations for the authors:**

**Reviewer #1 (Recommendations For The Authors):**
The authors have an elegant series of manipulations that provide strong evidence for their hypotheses and conclusions. Their exploitation of a unique biological system amenable to imaging in the larval salivary gland is well-considered and well-performed. Most of the conclusions are supported by the data. I only have the concerns below.(1) One of the main findings is the composition of Notch nuclear complexes and their interactions within a 'hub'. Yet most of the data showing hubs focus on labeling one protein component (+the locus or transcription), but multi-color imaging is rarely used to show how CSL-Mam, Mam-Med... protein signals coalescence to form a hub. Given the powerful tool developed, it would be important to show these multi-state hubs. Related to this, if the authors expect that hubs are formed independently of transcription or Notch pathway activation, do the authors see clustering at other non-specific loci in the nucleus? If not, can the authors comment on why they think that is the case? If so, do they demonstrate consistent residence time profiles with the tracked E(spl) locus?

We apologise that it was not evident from the data shown that the proteins co-localize. First we stress that all the experiments are multicolor and most rely on very powerful methods to measure co-recruitment at a chromosomal locus- something that is very rarely achieved by others studying hubs. Second, we have in all cases confirmed that the proteins do colocalize. We have modified the diagram of our analysis pipeline to make more clear that this relies on multi-colour imaging, and adjusted all the figure labels to indicate the position of E(spl)-C. We have also added panels to new supplementary Figure S1C with examples of the co-localization between CSL and Mam and a plot confirming their levels of recruitment are correlated across multiple nuclei.

We would like to clarify that our data show that the hubs do require Notch activation for their establishment. Other regions of enrichment are detected in Notch-ON conditions, but these are less prominent and, with no independent method for identifying them, can’t be compared between nuclei. In SPT experiments, other clusters with consistent residence are detected as reported in our recent paper which expanded on the SPT data (Baloul et al, 2023). We also detect co-localizations and “hubs” in other tissues, but those analyses are ongoing and beyond the scope of this paper.

(2) The authors convincingly show that Notch hub complexes exhibit a memory. While the data showing rapid hub reformation upon Notch withdrawal are solid and convincing (Figure 5, in particular, F), the claim that this memory fosters rapid transcriptional reactivation is less clear. Yet in order to invoke transcriptional memory, it's necessary to solidify this transcriptional response angle. The authors should consider quantifying the changes in transcription activity (at the TS and not in the cytoplasm as currently shown), as well as the timing of transcriptional reactivation (with the MS2 system or smFISH). Manipulating the duration of the activation and dark recovery periods could help to draw a better correlation between the timing of hub reformation and that of transcriptional response and would also help determine how persistent this phenomenon is.

We thank the reviewer for these suggestions. We have carried out several new experiments to probe further the persistence of memory and to show the effects on transcription when Notch is inactivated/reactivated. First, we have extended the time period for Notch inactivation by temperature control and show that the CSL hub persists even at 24 hours and that no transcription from the target E(spl)m3 is detected –neither at the transcription start-site nor in the cytoplasm. Second, we have extended the Notch OFF time period to 24 hours using the optogenetic approach and show that transcription is robustly reinitiated in preactivated nuclei when Notch is re-activated with 30 mins light treatment while little if any E(spl)m3 transcription is detected in naïve nuclei with the same treatment. These new data are included in new Figure 5 C-F and see page 13-14. Both these new experiments substantiate the model that the nuclei retain transcriptional memory.

(3) The manuscript ends with the finding that the presence of a Mam hub does not always correlate with transcription. They conclude that transcription is initially stochastic. The authors find this surprising and even state that this could not be observed without their in vivo live imaging approaches. I don't understand why this result is surprising or unexpected, as we now know that transcription is generally a stochastic process and that most (if not all) loci are transcribed in a bursting manner. The fact that E(spl)-C locus is bursty is already obvious from the smFISH data. The fact that active nascent transcription does not correlate with local TF hubs was already observed in early *Drosophila* embryos (with Zelda hubs and two MS2 reporters, hb-MS2, sna-MS2). If, in spite of the inherent stochasticity of transcription (bursting), the data are surprising for other reasons, the authors should explain it better.

We apologise that we had not made clear the reasons why the results were unexpected. We have substantially rewritten this section, and the discussion section, to clarify. We have also moderated the language used to better reflect the overall context of our results. We briefly summarise here. As the reviewer correctly states, it is well known that transcription is inherently bursty. Indeed the MS2 transcription profiles in “ON” nuclei are bursty, which likely reflects the switching of the promoter. However, in other contexts where we have monitored transcription although it is bursty it has nevertheless been initiated synchronously in response to Notch in all nuclei in a manner that was fully penetrant. What we observe in our current conditions, is that some nuclei never initiate transcription over the time-course of our experiments (2-3 hours), and those that are ON rarely switch off. This implies that there is another rate-limiting step. Supplying a second signal can modulate this so that it occurs with much higher frequency/penetrance. We consider this to be a second tier of regulation above the fundamental transcriptional bursting.

The fact that Mam is recruited in all nuclei, whether or not they are actively transcribing was surprising because recruitment of the activation complex has been considered as the limiting step. This is somewhat different from Zelda, which is thought to be permissive and needed at an early step to prime genes for later activation rather than to be the last step needed to fire transcription. We note also that we are not monitoring the position of the hub with respect to the promoter, as in the Zelda experiments (Zelda hubs may still persist, but they are not overlapping with the nascent RNA), we are monitoring the presence or absence of Mam hub in proximity to a genomic region.

Minor suggestions:(1) The genotypes of the samples should be indicated in the figure legends.

We thank the reviewer for this suggestion. We have provided a table (new Table S3) where all of the genetic combinations are provided in detail for each figure. We considered that this approach would be preferable because it would be quite cumbersome to have the genotypes in each legend as they would become very long and repetitive.

(2) While the schematic Fig1A explains how the locus is detected, the presence of ParS/ParB is never indicated in subsequent panels and Figure. I assume that all panels depicting enrichment profiles, use a given radius from the ParS/ParB dot to determine the zero of the x-axis (grey zone). This should be clearly stated in all panels/figure legends concerned.

We apologies if this was not made explicit. Yes, all panels depicting enrichment profiles, use immunofluorescence signal from ParA/ParB recruitment to determine the zero of the x-axis. We have now marked this more clearly In all figures (grey bar, grey shading or labelled 0). All images where the locus is indicated by an arrowhead, by a coloured bar above the intensity plots or by grey shading in the graphs have been captured with dual colour and the signal from ParA/B recruitment used to define its location. This is now clearly stated in the analysis methods and in the legend. We have also modified the diagram in new supplementary Figure S1B, showing our analysis pipeline, to make that more explicit.

(3) FRAP/SPT experiments: the author should provide more details. How many traces? Are traces showing bleaching removed?P7: does the statement ' The residences are likely an underestimation because bleaching and other technical limitations also affect track durations' imply that traces showing bleaching have not been removed from the analysis?The authors could justify the choice of the model for fitting FRAP/Spt experiments and be cautious about their interpretation. For example, interpreting a kinetic behavior as a DNA-specific binding event can be accurate, only if backed up with measurements with a mutant version of the DNA binding domain.

We apologise if some of this information was not evident. The number of trajectories is provided in new Figure S1F, which indicates the number of trajectories analyzed for each condition in Figure 1.

We have now added also the numbers of trajectories analyzed for the ring experiments.

The comments on page 7 about bleaching refer to the technical limitations of the SPT approach. However, as bleached particles cannot be distinguished from those that leave the plane of imaging, they have not been filtered or removed. We have not sought to make claims about absolute residence times for that reason. Rather the point is to make a comparison between the different molecules. As the same fluorescent ligand and imaging conditions are used in all the experiments, all the samples are equivalently affected by bleaching. We subdivide trajectories according to their properties and infer that those which are essentially stationary are bound to chromatin, as is common practice in the field. We note that we have previously shown that a DNA binding mutant of CSL does not produce a hub at E(spl)-C in Notch-ON conditions and has a markedly more rapid recovery in FRAP experiments (Gomez-Lamarca et al, 2018) consistent with the slow recovery being related to DNA binding. This point has been added to the text (page 8).

(4) The authors should quantify their RNAi efficiency for Hairless-RNAi, Med13-RNAi, white-RNAi, yellow-RNAi, CBP-RNAi, and CDK8-RNAi.

We thank the reviewer for this comment. We have made sure that we are using well validated RNAis in all our experiments and have included the references in Table S2 where they have been used. We have now evaluated the knock-down in the precise conditions used in our experiments by quantitative RT-PCR and added those data, which show efficient knock-down is occurring, to new Supplementary Figure S1D and Figure S3J. We note also that the RNAi experiments are complemented by experiments inhibiting the complexes with specific drugs and that these yield similar results.

(5) Figure 3 A: could the author show that transcription is indeed inhibited upon triptolide treatment with smFISH (with for example m3 probes)? Why not use alpha-amanitin?

We thank the reviewer for this suggestion. We had omitted the smFISH data from this experiment in error. These data have now been added to new Supplementary Figure S3A and clearly show that transcription is inhibited following 1 hour exposure to triptolide. Triptolide is a very fast acting and very efficient inhibitor of transcription that acts at a very early step in transcription initiation. In our experience it is much more efficient than alpha-amanitin and is now the inhibitor of choice in many transcription studies.

(6) Figure 4 typo: panel B should be D and vice versa. Accessibility panels are referred to as Figure 4D, D' in the text but presented as panel B in the Figure.

We thank the reviewer for noting this mistake, it is now changed in the main text.

(7) The authors must add their optogenetic manipulation protocol to their methods section.

The method is described in detail in a recently published paper that reports its design and use. We have now also added a section explaining the paradigm in the methods (Page 31) as requested.

(8) Figure 3G needs a Y-axis label.

Our apologies, this has now been added.

(9) The authors should note why there was a change of control in Figure 3D compared to 3E and G (yellow RNAi vs white RNAi).

This is a pragmatic choice that relates to the chromosomal site of the RNAis being tested. Controls were chosen according to the chromosome that carries the UAS-RNAi: for the second chromosome this was yellow RNAi and for the third white RNAi. This is explained in the methods.

(10) Figure 1 would benefit from a diagram describing the genomic structure of the E(spl) locus and the relative position of the labelled locus within it.

We thank the reviewer for this suggestion and have added a diagram to Supplementary Figure S1A .

**Reviewer #2 (Recommendations For The Authors):**
Minor criticisms and typos:Pet peeve: in some of the figure panels they are labeled Notch ON or OFF, but in others they are not, albeit that info is included in the figure legend. For the ease of the reader/reviewer, would it be possible to label all relevant figure panels either Notch ON or OFF for clarity?

We thank the reviewer for this suggestion and have modified the figures accordingly.

Page 7, top. "In comparison to their average distribution across the nucleus, both CSL and Mam trajectories were significantly enriched in a region of approximately 0.5 μm around the target locus in Notch-ON conditions, reflecting robust Notch dependant recruitment to this gene complex." Are the authors referring to Figure 1D here?

Thank you, this figure call-out has been added in the text.

Page 9. "...reported to interact with p300 and other factors (Figure S2B)." I believe the authors mean Figure S2C and not S2B.

Thank you, this has been corrected in the text.

Page 9. There is no Figure S2D.

Apologies, this was referring to Figure S1D, and is now corrected in the text.

Page 11: "...were at very reduced levels in nuclei co-expressing MamDN (Figure 4B).." Should be Figure 4CD.

Thank you, this has been corrected in the text.

Page 12: "...which was maintained in the presence of MamDN (Figure 4D, D')." Should be Figure 4B.

Thank you, this has been corrected in the text.

**Reviewer #3 (Recommendations For The Authors):**
In the Results section on Hub, the paragraph starting with "Third, we reasoned . ." the callout to Figure S2D should be Fig S1D.

Thank you, this has been corrected in the text

Figures: The font size in the Figures is so small that most words and numbers cannot be read on a printout. One has to go to the electronic version and increase the size to read it. This reviewer found that inconvenient and often annoying.

We apologise for this oversight, the font size has now been adjusted on all the graphs etc.

Figure legends: the legends are terse and in some cases leave explanations to the imagination (e.g. "px" in Figure 2E). It would be useful to go through them and make sure those who are not a *Drosophila* Notch person and not a transcription biochemist can make sense of them.

Our apologies for the lack of clarity in the legends. We have gone over them to make them more accessible and less succinct.